# Circumglobal Rossby wave patterns during boreal winter highlighted by wavenumber/phase speed spectral analysis

Jacopo Riboldi[1,3], Efi Rousi[2], Fabio D'Andrea[1], Gwendal Rivière[1], and François Lott[1]

[1]Laboratoire de Météorologie Dynamique, École Normale Supérieure, PSL University, Paris, France
[2]Potsdam Institute for Climate Impact Research (PIK), Leibniz Association, Potsdam, Germany
[3]Department of Earth Sciences, Uppsala University, Uppsala, Sweden

**Correspondence:** Jacopo Riboldi (jacopo.riboldi@geo.uu.se)

**Abstract.** The classic partitioning between slow-moving, low-wavenumber planetary waves and fast-moving, high-wavenumber synoptic waves is systematically extended by means of a space/time spectral decomposition to characterize the day-to-day evolution of Rossby wave activity in the upper troposphere. This technique is employed to study the origin and the propagation of circumglobal Rossby wave patterns (CRWPs), amplified Rossby waves stretching across the Northern Hemisphere in the zonal direction and projecting primarily over few, dominant wavenumber/phase speed harmonics. Principal component analysis of daily anomalies in spectral power allows for two CRWPs to emerge as leading variability modes in the spectral domain during boreal winter. These modes correspond to the baroclinic propagation of Rossby wave packets (RWPs) from the Pacific to the Atlantic storm track in a hemispheric flow configuration displaying enhanced meridional gradients of geopotential height over midlatitudes. The first CRWP is forced by tropical convection anomalies over the Indian Ocean and features the propagation of amplified RWPs over northern midlatitudes, while the second one propagates rapidly over latitudes between 35°N and 55°N and appears to have extratropical origin. An anomalous equatorward propagation of Rossby waves from the Atlantic eddy-driven jet to the North African subtropical jet is observed for both CRWPs. The obtained results highlight the substantial contribution of propagating RWPs to CRWPs, hinting that the two features might have the same nature.

## 1 Introduction

Most of the weather systems affecting northern midlatitudes have their dynamical origin in the Atlantic and Pacific storm track regions, located to the east of the main continental landmasses and characterized by a particularly strong meridional temperature gradient (e.g., Chang et al., 2002; Hakim, 2003). The majority of extratropical cyclones form and track across those regions while being steered by large-scale atmospheric oscillations, known as Rossby waves, constituted by equatorward (troughs) and poleward (ridges) displacements of the jet stream (Rossby, 1940; Lee and Held, 1993; Wirth et al., 2018). The position and the activity of the Pacific and Atlantic storm tracks, in terms of number, amplitude and propagation of Rossby waves populating them, can vary substantially across seasons and between different years (Grise et al., 2013; Hoskins and Hodges, 2019a, b).

Understanding the mechanisms governing this variability is a traditional field of research in weather and climate dynamics. A possible division of this research area could involve a distinction between local mechanisms, at the level of a single storm track,

and hemispheric mechanisms, featuring a coordination between the two. An example of local mechanism is the complex role played by background baroclinicity in enhancing, but also in suppressing, storm track activity over the North Pacific (Schemm and Schneider, 2018; Schemm et al., 2021). The onset of atmospheric blocking events, characterized by the sudden inflation of large anticyclones that disrupt the usual, eastward Rossby wave propagation, can also be seen as a local mechanism affecting storm track variability (e.g., Shutts, 1983; Altenhoff et al., 2008; Martineau et al., 2017; Ferranti et al., 2018). On the other hand, several hemispheric mechanisms connecting the two storm tracks have been identified. There is robust evidence that low-frequency tropical forcing, as the El Nino-Southern Oscillation (ENSO) or the Madden-Julian Oscillation (MJO), manifests itself first in anomalies over the North Pacific storm track that are subsequently propagated, via the forcing of transient Rossby waves, to the North Atlantic storm track (Cassou, 2008; Henderson et al., 2016; Zheng et al., 2018; Schemm et al., 2018; Gollan et al., 2019). However, interaction between the two storm tracks can in principle result from purely extratropical forcing, for instance in the form of baroclinically-induced wave resonance (Yang et al., 1997; Franzke et al., 2006). An interplay between storm tracks is also related to hemispheric-scale, extratropical teleconnection patterns like the Northern Annular Mode (e.g., Woollings and Hoskins, 2008; Rivière and Drouard, 2015) or the Northern Baroclinic Annular Mode (Thompson and Li, 2015).

Other circulation patterns extending beyond the scale of a single storm track are the so-called circumglobal Rossby wave patterns (CRWPs), observed when a single Rossby wave extends around a substantial portion of a latitude band (White et al., 2021). The existence of CRWPs and their differences from more usual Rossby wave packets (RWPs) is an open research question, fueled by the potential role of CRWPs in causing extreme weather events. Concomitant summer temperature extremes in different regions of the Northern Hemisphere have been linked to the occurrence of slow-moving, amplified, quasi-resonant planetary Rossby waves (Petoukhov et al., 2013; Coumou et al., 2014; Kornhuber et al., 2017, 2019, 2020; Xu et al., 2021). During boreal winter, on the other hand, CRWPs stretching across the hemisphere feature an interconnection between the Pacific and the Atlantic storm track and an interplay between the polar and the subtropical jet streams (Branstator, 2002; Franzke et al., 2006; Davies, 2015; Harnik et al., 2016). In particular, Davies (2015) described how the repeated propagation of a Rossby wave train from the Pacific to the Atlantic storm track, and then back to the Pacific storm track via the subtropical jet stream, shaped the hemispheric wave pattern during winter 2013/14 causing widespread extreme weather. This so-called "weather chain" involved the repeated enhancement of RWPs with the same phase (Röthlisberger et al., 2019) and an interaction between the polar and subtropical jet to ensure circumglobal connectivity, and occurred in conjunction with anomalous North Pacific sea surface temperatures (Wang et al., 2014; Hartmann, 2015). In contrast with previous studies, Davies (2015) highlighted the contribution of several propagating RWPs to the appearance of the CRWP, by tracing them in a Hovmöller diagram.

Despite these research efforts, substantial theoretical and methodological challenges still hinder progress towards shared definitions and methodologies to study CRWPs (White et al., 2021). The main issues concern the diagnosis of circumglobal waveguides and the linkages between CRWPs and more classical RWPs. The theory of quasi-resonant amplification for Rossby waves, developed by Petoukhov et al. (2013) and employed in many successive studies to diagnose CRWPs, relies on Rossby wave ray tracing (cf. Hoskins and Karoly, 1981; Hoskins and Ambrizzi, 1993), an approach based on the Wentzel-Kramers-Brillouin (WKB) approximation. However, recent research work has discussed the limits of these theoretical foundations

when applied to evaluate the presence of hemispheric-scale waveguides (Wirth, 2020; Wirth and Polster, 2021). From the methodological point of view, many studies identify CRWPs in time-averaged fields of upper-level wind or geopotential height, usually employing weekly (Kornhuber et al., 2019), bi-weekly (Kornhuber et al., 2017) or monthly means (e.g., Branstator, 2002; Petoukhov et al., 2013). The drawback of this approach is that temporal averaging acts implicitly as a low-pass filter and excludes RWPs and rapidly propagating transients from the analysis: this can lead to problems when trying to diagnose CRWPs and circumglobal waveguides (as discussed by Fragkoulidis et al., 2018). An example of this issue appears in the topic of recurrent Rossby wave packets, where the repeated genesis and eastward propagation of RWPs with similar longitudinal phase is observed (e.g., Röthlisberger et al., 2019). Those propagating transients would appear as a stationary Rossby wave if a time mean were to be taken, leading to a potential misunderstanding of the nature of the phenomenon (cf. Ali et al., 2021, their Fig. 1). A similar confusion could in principle occur with CRWPs, as it is not fully clear how much of their time-averaged signature results from the circumglobal propagation of RWPs rather than from mechanisms involving planetary waves. An additional issue concerns the diagnosis of a single, dominant wavenumber along a latitude circle to characterize CRWPs (e.g., wave-5 versus wave-7), because non-circumglobal, disconnected RWPs could also feature a clear spectral projection on the same wavenumbers.

This study consists of an exploration of northern hemispheric Rossby wave variability during boreal winter, with a particular focus on CRWPs. The usual spectrum of zonal wavenumbers describing the hemispheric flow is complemented here by the inclusion of the temporal dimension in the spectral analysis, that allows to study the zonal propagation of Rossby waves over a given period of time. This technique permits to explicitly separate the contributions of slow and fast waves for each zonal wavenumber, without the need of filtering. The first objective of this study is to show that space/time spectral analysis can indicate the presence of CRWPs. We assume that a CRWP, if it is occurring, would project on a well-defined range of zonal wavenumbers and phase speeds as picked up by the spectral decomposition. This indirect approach allows to assess the presence of CRWPs with as few as possible a-priori assumptions about their nature or without having to explicitly diagnose the presence of waveguides, making the results less dependent on subjective choices. We show that hemispheric-scale Rossby wave patterns dominated by few, selected wavenumber/phase speed harmonics appear as basic modes of variability in the spectral domain, and that such patterns are associated with CRWPs. This first conclusion leads to the second objective, that is to describe some dynamical features of the origin and propagation of the identified CRWPs, and their connection with RWPs. The paper is structured as follows: after a description of the spectral analysis and of the employed diagnostics, the patterns of spectral variability and their associated large-scale circulation signatures are discussed in general (Sec. 3) before focusing specifically on the characteristics of the CRWPs highlighted by the spectral analysis (Sec. 4). Some implications of the results for CRWP research, as well as a few limitations of the employed method, are discussed in Sec. 5. Conclusions are summarized in Sec. 6, together with an outlook about possible applications of space/time spectral analysis in the context of climate dynamics.

## 2 Data and methods

The analysis is based on ERA-Interim Reanalysis data (Dee et al., 2011) between March 1979 and February 2019, for a total length of 40 years. Spectral decomposition is performed using a rapid Fortran-based fast Fourier transform routine, developed in 1967 at MIT Lincoln Laboratory by Norman Brenner, Charles Rader and Ralph Alter (Cooley and Tukey, 1965). Daily MJO data for the considered period are obtained from the web page of the Australian Bureau of Meteorology (http://www.bom.gov.au/climate/mjo).

### 2.1 Wavenumber/phase speed spectral analysis

Spectral analysis allows a decomposition of a given function over a basis of periodic harmonics: in this work, each harmonic corresponds to a sinusoidal Rossby wave of zonal wavenumber $n$ and propagating eastward with phase speed $c_p$. These harmonics are obtained from an interpolation, performed at each latitude $\phi$, of the wavenumber/frequency harmonics $(n, \omega)$ resulting from space/time spectral analysis. The technique is employed routinely to diagnose equatorial waves and other circulations existing in the tropical region (e.g., Wheeler and Kiladis, 1999; Lott et al., 2009). Applications to the extratropics have mainly described the climatological characteristics of Rossby waves or their differing representation in climate models (e.g., Dell'Aquila et al., 2005). Recent work (Sussman et al., 2020; Riboldi et al., 2020; Jolly et al., 2021) employed space/time spectral analysis to model and assess the presence of trends in the properties of Rossby waves, confirming its usefulness in tackling research questions in climate dynamics.

The employed procedure, detailed in the following paragraphs, is the same as in Riboldi et al. (2020). In the first step a double spectral analysis, in space and time, is performed over the ERA-Interim data set of 250 hPa meridional wind anomalies $V'(\lambda, t; \phi)$, with six-hourly resolution between January 1979 and December 2019 and spatial resolution of $0.75° \times 0.75°$. Such anomalies are computed with respect to the annual cycle of meridional wind, further smoothed using a 30-day moving average. The employed temporal and spatial resolution can resolve harmonics of minimal zonal wavelength of $1.5°$ and minimal frequencies corresponding to an oscillation period of 12 h. This level of detail allows a precise depiction of Rossby waves associated with rapid transients like, e.g., moving extratropical cyclones (Wheeler and Kiladis, 1999; Dell'Aquila et al., 2005).

The flow is decomposed on each latitude circle as a linear superposition of monochromatic, zonally propagating waves

$$V'(\lambda, t; \phi) = \sum_{j=-N_T/2}^{N_T/2} \sum_{n=-N_L/2}^{N_L/2} \hat{V}'(n, \omega_j; \phi) e^{i(n\lambda - \omega_j t)} \tag{1}$$

where $\hat{V}'(n, \omega_j; \phi) \in \mathbb{C}$ are the spectral coefficients, $\lambda$ represents longitude in radians (from 0 to $2\pi$) and $t$ is time, $N_L = 480$ is the number of grid points along a given latitude circle on the considered grid and $\omega_j = 2\pi j / N_T$ is the frequency, ranging in a time window of 61 days (containing $N_T = 244$ six-hourly time steps). The spectral analysis was performed in a sliding two-month time window centered at 12UTC of each day between the $1^{st}$ of February 1979 and the $30^{th}$ of November 2019. A double cosine tapering is applied to the first and last 12 days of the time series to minimize Gibbs effect. The modules $P(n, \omega_j, \phi) = \hat{V}'\hat{V}'^*$ of the complex coefficients obtained from the spectral analysis are an estimate of the density of spectral power and constitute the so-called periodogram. By Parseval's identity, each value of the periodogram is proportional to the

variance of the meridional wind anomalies explained by the corresponding $(n, \omega_j)$ harmonic. Each periodogram is smoothed 10 times in frequency with a 1-2-1 running average to further reduce noise (Wheeler and Kiladis, 1999; Small et al., 2014).

The periodogram $P(n, \omega_j; \phi)$ is then linearly interpolated along lines of constant phase speed $c_p = \frac{\omega a \cos \phi}{n}$, where $a$=6.371·$10^6$ m is the Earth's radius, to obtain periodograms in the wavenumber/phase speed domain $P(n, c_p; \phi)$, following the approach described in Randel and Held (1991). The interpolation is performed on a range of phase speeds between -30 m s$^{-1}$ and +30 m s$^{-1}$ in steps of $\Delta c_p$ =1 m s$^{-1}$. The periodograms are multiplied by a wavenumber- and latitude-dependent scaling factor $\frac{n}{a \cos \phi}$ before interpolation to preserve variance during the transformation from the frequency to the phase speed dimension (cf. Eq. 3b

in Randel and Held, 1991). A detailed explanation of the interpolation procedure is given in the Supplement (Text S1). The choice of the phase speed range implies that harmonics with phase speed $|c_p|$ <1 m s$^{-1}$ or $|c_p|$ >30 m s$^{-1}$ at each latitude are excluded, as already noticed by Randel and Held (1991). This latitudinal dependence partly limits the harmonics that the spectral analysis can resolve, with a more pronounced effect for quasi-stationary, low-wavenumber harmonics at low latitudes: a detailed description of this effect is provided in the Supplement (Text S2). Harmonics with $c_p$=0 m s$^{-1}$ are also not represented,

because they correspond to waves with null frequency and, therefore, infinite period.

As a final step, a global estimate $\overline{S}(n, c_p)$ of the Rossby wave spectrum over midlatitudes is obtained by summing the wavenumber/phase speed periodograms computed for each latitude between 35.25°N and 75°N (every 0.75°). The integration across different latitudes acts as an additional "smoother" and allows to obtain less noisy spectra. This broad latitudinal range has been chosen 1) to increase the range of resolved wavenumber/phase speed harmonics, which is latitude-dependent (as

detailed in Text S2 of the Supplement) and 2) to account for intraseasonal and interannual latitudinal variations in Rossby wave propagation and storm track activity. Although a spectrum is attributed to every single day, it is important to state that each spectrum results from an analysis over a time interval of 61 days (37 if we exclude the tapered days), with a substantial overlap between consecutive days. Despite this apparent limitation, however, the results outlined in this work indicate that the methodology is able to sample a notable intraseasonal variability and to precisely pinpoint time periods of anomalous spectral

activity at the daily to weekly scale. An alternative approach would have consisted in performing first the latitudinal average of the wind anomalies and then a single spectral analysis, as in Dell'Aquila et al. (2005): however, meridional wind anomalies can sometimes have opposite signs at different latitudes for the same longitude, which would lead to the cancellation of the signal. Configurations of Rossby wave breaking and atmospheric blocking are two examples where this effect might be particularly important: indeed, some blocking identification methods rely precisely on this meridional superposition of anomalies (e.g.,

Tibaldi and Molteni, 1990; Pelly and Hoskins, 2003; Davini et al., 2012).

All the daily spectra can be averaged to obtain seasonal mean spectra, highlighting some basic characteristics as well as the seasonal cycle of $\overline{S}(n, c_p)$ across the different harmonics (Fig. 1). The seasonal evolution of wavenumbers and phase speeds across seasons is portrayed by the metrics $[n]$ and $[c]$, respectively: they are obtained by weighting $n$ and $c_p$ with respect to $\overline{S}(n, c_p)$, as in Riboldi et al. (2020). As intuitively expected, the average spectra are not symmetric: more power is present for

eastward-propagating waves than for westward ones, with a maximum at wavenumbers $n$=5-7 depending on the season. High wavenumbers tend to exhibit more power in harmonics with high phase speed as predicted by the Rossby wave dispersion relationship. However, westward-propagating waves are also active for most wavenumbers below $n$ =8, and they reach their

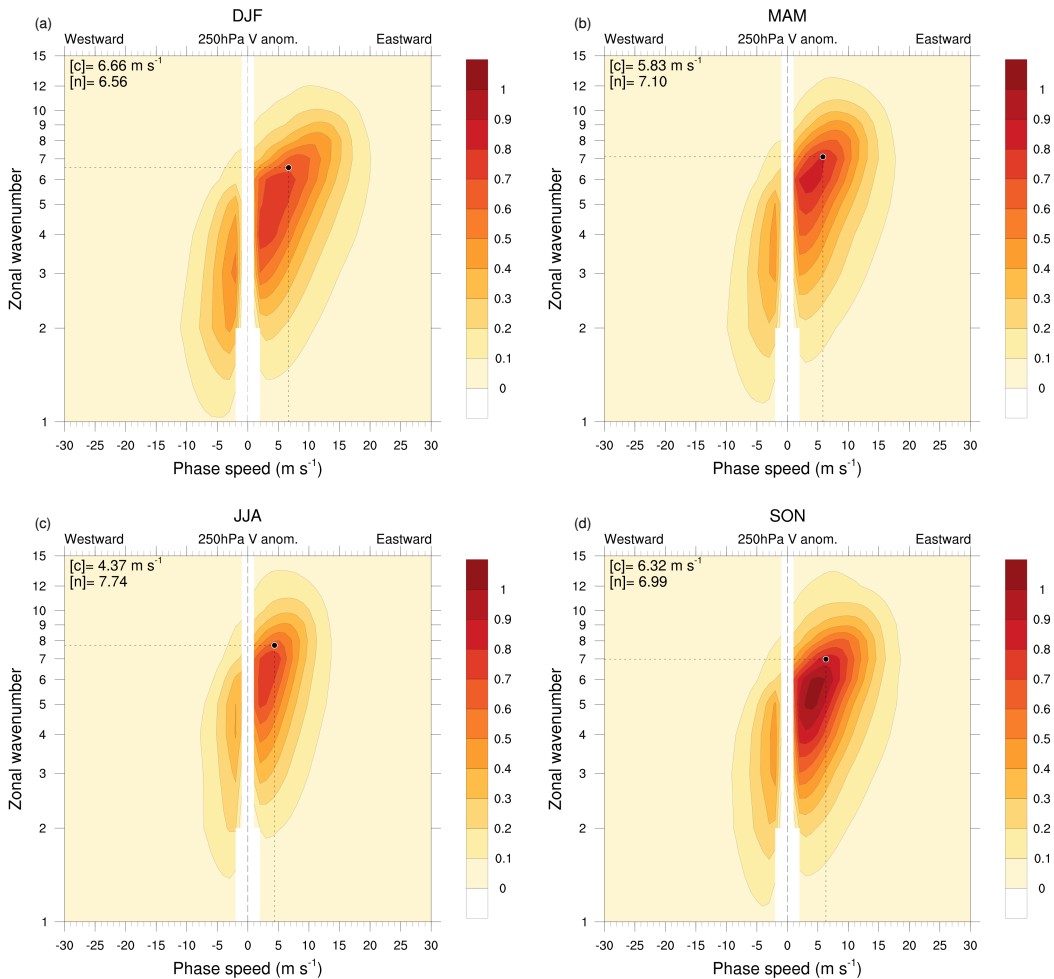

**Figure 1.** Mean of daily wavenumber/phase speed spectra of meridional wind anomaly at $250\,\mathrm{hPa}$, computed along each latitude circle between $35.25°\mathrm{N}$ and $75°\mathrm{N}$, every $0.75°\mathrm{N}$, during the four seasons: (a) DJF, (b) MAM, (c) JJA and (d) SON. Units of spectra is $\mathrm{m^2\,s^{-2}\,\Delta c^{-1}}$, where $\Delta c = 1\,\mathrm{m\,s^{-1}}$ at $55°\mathrm{N}$, and values of $\overline{S}(n, c_p)$ are multiplied by $n$ because of the logarithmic scale adopted for the wavenumber axis. Black dots indicate the phase speed $[c]$, as in Riboldi et al. (2020), and the average wavenumber $[n]$: , both expressed as a weighted averages with respect to spectral power. The respective values are reported in the top left corner of each plot.

maximum power during winter. We speculate that the power in those retrogressive harmonics is due to the occurrence of atmospheric blocking and Rossby wave breaking, phenomena that feature a disruption of the normal eastward propagation of upper-level troughs and ridges. The distribution of $\overline{S}(n, c_p)$ is flattened around the $c_p=0\,\mathrm{m\,s^{-1}}$ line during summer: this indicates that the Rossby wave pattern governing weather conditions over Northern Hemisphere midlatitudes is, on average, more stationary during summer than in winter. The involved wavenumbers are also generally higher during summer, with intermediate values across the transition seasons (MAM and SON). Although spectra have been computed across all seasons, we will only focus on boreal winter (DJF) through the rest of the paper.

## 2.2 Dynamical diagnostics

The circulation patterns associated to modes of spectral variability are described making use of standard meteorological variables such as potential vorticity (PV), geopotential height and zonal wind. More elaborated diagnostics are also employed in the paper, as atmospheric blocking frequency, baroclinicity and transient meridional heat flux. The RWP diagnostic by Zimin et al. (2006) and the $\mathbf{E}$ vector by Hoskins et al. (1983) are also used to stud amplitude and propagation of the RWPs associated with the identified CRWPs. These less straightforward metrics are described in the following paragraphs.

### 2.2.1 Atmospheric blocking

The anomaly-based blocking diagnostic by Schwierz et al. (2004) is employed in this study. In this framework, atmospheric blocking corresponds to a negative anomaly of vertically averaged PV (500-150 hPa, anomalies computed with respect to monthly climatology). After smoothing with a two-day running mean, blocking is identified as a closed region with a negative PV anomaly lower than -1.3 PV units (PVU, $1\,\mathrm{PVU} = 10^{-6}\,\mathrm{K\,m^2\,kg^{-1}\,s^{-1}}$). Furthermore, this region must fulfill an overlap criterion: the minimum spatial overlap of two closed regions delimited in two subsequent 6-hourly time steps must be 70% for at least 5 consecutive days. The output of this diagnostic is a two-dimensional boolean (0-1) field: composites of blocking frequency can be computed by determining the percentage of time steps where each grid point was occupied by a blocking. Anomalies in blocking frequency value are then obtained by subtracting the climatological blocking frequency, evaluated over all considered winter days between December 1979 and February 2019.

### 2.2.2 Rossby wave packet amplitude and propagation

The amplitude of Rossby wave packets in the wavenumber range $n = 4-15$ is evaluated at each grid point using the diagnostics by Zimin et al. (2006). To account for the possible non-zonal propagation of Rossby waves, the 250 hPa wind component orthogonal to a local approximation of the background flow (here, a streamline of the mean wind averaged in a time window of 28 days centered at the date of interest) is employed in the computation. This orthogonal wind component is first tapered in space using a Gaussian filter centered at the grid point of interest and then filtered using a Fourier transform to retain only the considered wavenumbers. The modulus of the inverse Fourier transform of the filtered wind is then the local amplitude of the identified Rossby wave packet envelope. As it represents an amplitude, this metric is positive definite and is not affected

by cancellation problems, due to phase shifts between ridges and troughs in the wave packet, when averaging different time steps together. As noticed by Riboldi et al. (2019), this metric does not follow a Gaussian distribution: standardized anomalies of RWP amplitude are then computed with respect to the amplitude logarithm.

The propagation and the tilt of transient eddies at the level of the jet stream are diagnosed using the **E** vector (Hoskins et al., 1983; Trenberth, 1986) in the formulation by Schemm et al. (2018)

$$\mathbf{E} = (E_x, E_y) = \left[ \frac{1}{2} \left( \overline{v^{*2} - u^{*2}} \right), -\overline{u^* v^*} \right] \tag{2}$$

where the transient 250 hPa wind components $u^*$ and $v^*$ are again defined with respect to 7-day running averages. The two components of the **E** vector for that same day are computed every 6 hours and then averaged to obtain a daily mean value (as indicated by the overbar). The horizontal component $E_x$ is linked to the group speed of transient eddies and can be used to diagnose the preferred direction of propagation of wave energy (Orlanski, 1998). The meridional component $E_y$ is linked to the tilt of transient eddies: equatorward-pointing **E** vectors indicate anticyclonically tilted eddies, while poleward-pointing **E** vectors indicate a cyclonic tilt. Furthermore, as hinted by the similarities of $E_y$ with the eddy flux of zonal momentum, the divergence of **E** corresponds to transfer of zonal momentum from the transients to the mean flow.

### 2.2.3 Baroclinicity and transient meridional heat flux

The relationship of the identified spectral variability modes with local and global baroclinicity is assessed using the metric developed by Hoskins and Valdes (1990) and employed, among others, by Ambaum and Novak (2014) to study its relationship with meridional heat flux over the North Atlantic storm track.

The baroclinicity metric is essentially a version of the Eady growth rate for unstable baroclinic modes, evaluated at each grid point from the 7-day running averages of zonal wind ($\overline{u}$), meridional wind ($\overline{v}$) and potential temperature ($\overline{\theta}$):

$$s = 0.31 \frac{f}{\overline{N}} \left| \frac{d\overline{\mathbf{V}}}{dz} \right| \tag{3}$$

where $f$ is the Coriolis parameter,

$$\left| \frac{d\overline{\mathbf{V}}}{dz} \right| = \sqrt{\left( \frac{\partial \overline{u}}{\partial z} \right)^2 + \left( \frac{\partial \overline{v}}{\partial z} \right)^2} \tag{4}$$

is the average wind shear magnitude and

$$\overline{N} = \sqrt{\frac{g}{\overline{\theta}} \cdot \frac{d\overline{\theta}}{dz}} \tag{5}$$

the average stratification. Vertical derivatives are evaluated using centered finite differences between 700 hPa and 850 hPa, so that the value of $s$ refers to the 775 hPa level. Differently from Ambaum and Novak (2014), quantities averaged over 7 days have been employed to better represent the background baroclinicity in which transients are evolving.

The meridional heat flux associated with transient eddies ($v^*T^*$) is computed at each time step and grid point as the product of transient wind and temperature components at 700 hPa, both obtained by removing the corresponding 7-day running mean of wind and temperature. A seven-day running average is then applied to this instantaneous quantity to better compare it with the baroclinicity metric.

## 3 Variability in the spectral domain during boreal winter

The variability in the distribution of spectral power across the different $(n, c_p)$ harmonics is investigated with empirical orthogonal functions (EOFs) for boreal winter only (DJF). As a first step, spectral anomalies are computed with respect to the climatological seasonal cycle of spectral power. This is obtained by computing the average spectra for each calendar day and then by applying over it a 30-day running mean. Once the anomalies have been computed, standard Principal Component (PC) Analysis is performed over the 3610 winter days between December 1979 and February 2019. The resulting EOFs are consequently patterns of variability in the $(n, c_p)$ space, corresponding to the enhancement of selected harmonics with a given zonal scale and propagation.

### 3.1 EOF patterns

The three first EOFs explain together almost a third of the total variance in spectral space (31.56%, sum of respectively 12.54%, 9.97% and 9.05%); they are presented, together with the associated standardized PCs, in Fig. 2. The first EOF shows a compact maximum for the harmonics which already feature the highest average spectral power during DJF, the ones encompassing wavenumbers $n$ =4-6 and weakly positive phase speeds (Fig. 1a). Positive values of the associated principal component (PC1) correspond to a particular enhancement of such harmonics, while the opposite holds for negative values: we notice that the distribution of PC1 is skewed towards positive values, meaning that positive peaks have usually higher magnitude than negative ones (Fig. 2b). Eastward propagating $n$ =4-6 harmonics sit somehow in the middle between the classical planetary and synoptic wave subsets and have already been shown to be involved in circumglobal teleconnections during DJF (e.g., Branstator, 2002; Davies, 2015). The second EOF features a dipole in variability reminiscent of the distinction between planetary waves, having low wavenumber and phase speed, and synoptic waves, having high wavenumbers and rapidly propagating eastward (Fig. 2c). The positive phase of the associated PC2 features a distinct maximum in spectral power for zonal wavenumbers $n$ =6-8. The excited harmonics have also higher phase speeds than PC1, indicating an overall more rapid eastward propagation of the wave pattern. The third EOF exhibits a tripole of anomalies in the spectral domain, where harmonics having very low ($n$ =2, 3) and high ($n$ =6-8) wavenumber co-vary with an eastward propagating wave 5 (Fig. 2e). As for PC1, the distribution of PC3 is skewed to the right (Fig. 2f).

### 3.2 Circulation anomalies regressed from PCs

A regression analysis linking each PC with daily 250 hPa geopotential height, atmospheric blocking and RWP amplitude anomalies is performed to highlight the main circulation features associated with each spectral variability pattern. The positive

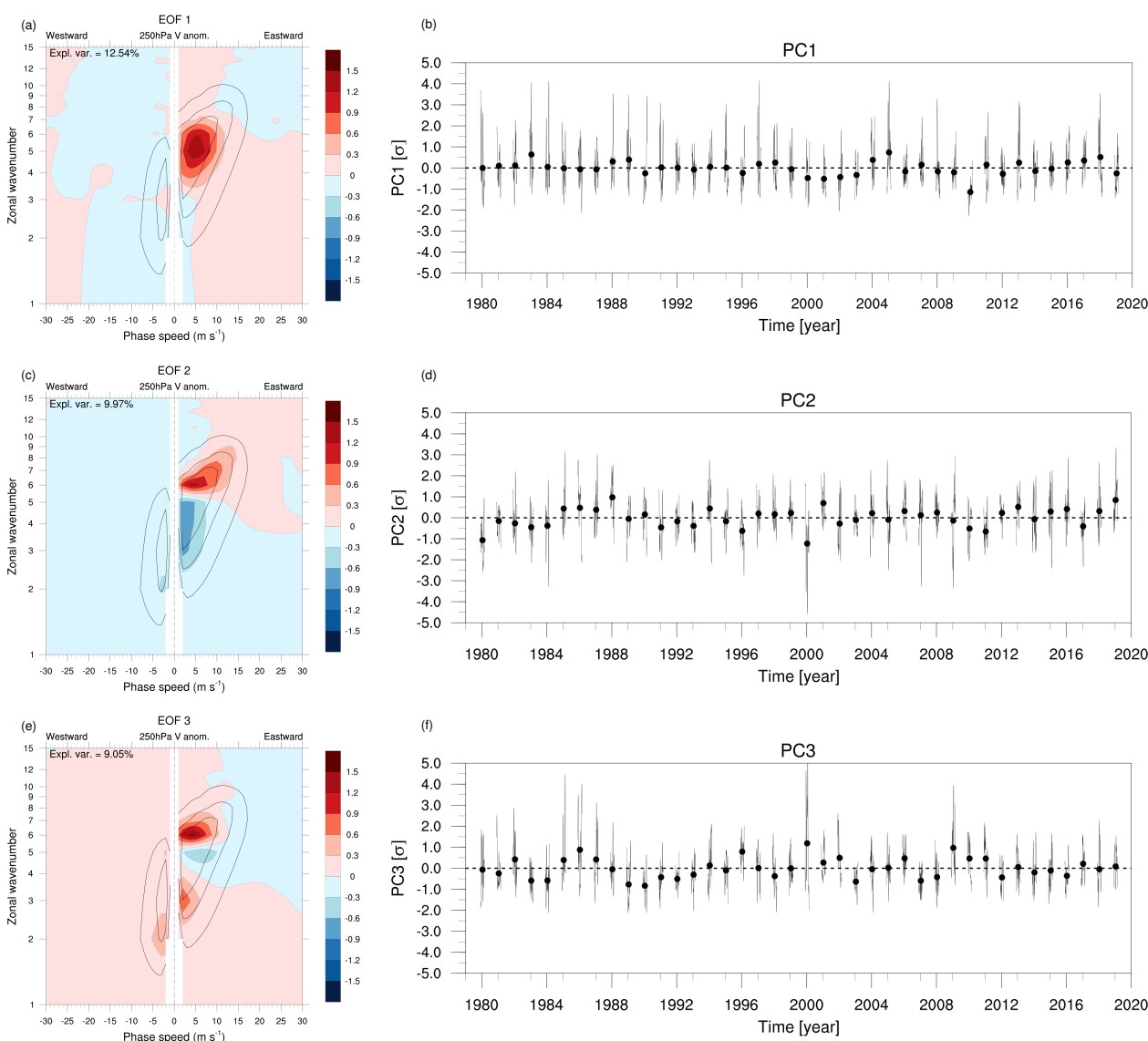

**Figure 2.** (a,c,e) Loadings of the first three EOFs of daily spectral anomaly during DJF (shaded, arbitrary units). The climatological spectral power is overlaid (black contours as in Fig. 2, only the 0.2, 0.4 and 0.6 $\mathrm{m\,s^{-1}\,\Delta c^{-1}}$ levels). (b,d,f) Principal components associated with each EOF (standardized).

phase of PC1 corresponds to an enhanced geopotential gradient over the Pacific and the Atlantic storm track, with a poleward-shifted jet stream (Fig. 3a). The higher latitude of the subtropical anticyclones is highlighted by the positive blocking frequency anomalies at lower latitudes than climatology (Fig. 3b). Notably, these blocks do not appear to be associated with a reversal of the meridional geopotential gradient over the midlatitudes, but instead with slow-moving ridges located between 35°N-55°N, detected as blocking by the Schwierz et al. (2004) identification algorithm (Fig. 3b). Perhaps the most remarkable feature associated with a positive PC1 phase is the simultaneous positive anomalies of RWP amplitude over the whole midlatitudes (Fig. 3c). These amplified Rossby waves are slowly propagating eastward, as they project on harmonics with $c_p < 10\,\mathrm{m\,s^{-1}}$.

The positive phase of PC2 features an overall increased meridional gradient of geopotential height at 250 hPa, which is also displayed by the positive zonal wind anomalies (Fig. 3d). It also features a reduced atmospheric blocking activity, in particular over the North Atlantic (Fig. 3e). The relationship of this PC with blocking is also proven by a significant anti-correlation with the daily time series of blocking area over the Northern Hemisphere ($r$=-0.53, $p < 0.01$). While PC1 is associated with a pulsation in RWP amplitude, PC2 seems more related to a latitudinal shift: positive PC2 values are related to enhanced RWP amplitude across midlatitudes, although at more southern latitudes than PC1, and with reduced amplitudes over high latitudes (Fig. 3f). The southward shift in RWP amplitude anomaly is likely related to the equatorward displacement of the Icelandic and Aleutian lows, possibly associated with the positive geopotential height anomalies above the North Pole.

The positive phase of PC3 is associated with a reduction of meridional geopotential gradient and with atmospheric blocking, in particular over the North Atlantic (Figs. 3g,h). Two separate regions of positive RWP amplitude anomalies are observed if PC3 is positive, one over high latitudes (north of 60°N) and one at lower latitudes (below 45°N). This split in RWP pathways suggests the presence of two separate Rossby waveguides (Figs. 3j).

## 4  Analysis of CRWP events

After having outlined the salient characteristics of the circulation patterns associated with the main modes of spectral variability, we shift our focus on CRWP events. We select events in the top and bottom 15% of each PC to study, from a composite perspective, some aspects of the origin and propagation of the associated CRWPs. The considered events have a minimum duration of five consecutive days and the minimum time separation between two consecutive events is at least of 10 days.

As a first step, we analyze the degree of zonal symmetry in the propagation of transient waves using the zonal component of the E-vector (Fig. 4). Events of particularly high PC1 and PC2 are associated with enhanced eastward propagation of transient waves, as well as with positive anomalies in RWP amplitude over a substantial portion of northern midlatitudes (Figs. 4a,c). These features, together with the peculiar projection over few, distinct $(n, c_p)$ harmonics in the corresponding EOFs, suggest that these events correspond to CRWPs. The other sets of events do not share such characteristics: RWP amplitudes are significantly reduced over a broad portion of midlatitudes during low PC1 events (Fig. 4b), while $E_x$ and RWP amplitude anomalies are absent or localized to smaller portions of the hemisphere for high/low PC3 or low PC2 events (Figs. 4d-f). The identified patterns do not change substantially if events in the top 20% of top 10% of the corresponding PCs are considered (not shown).

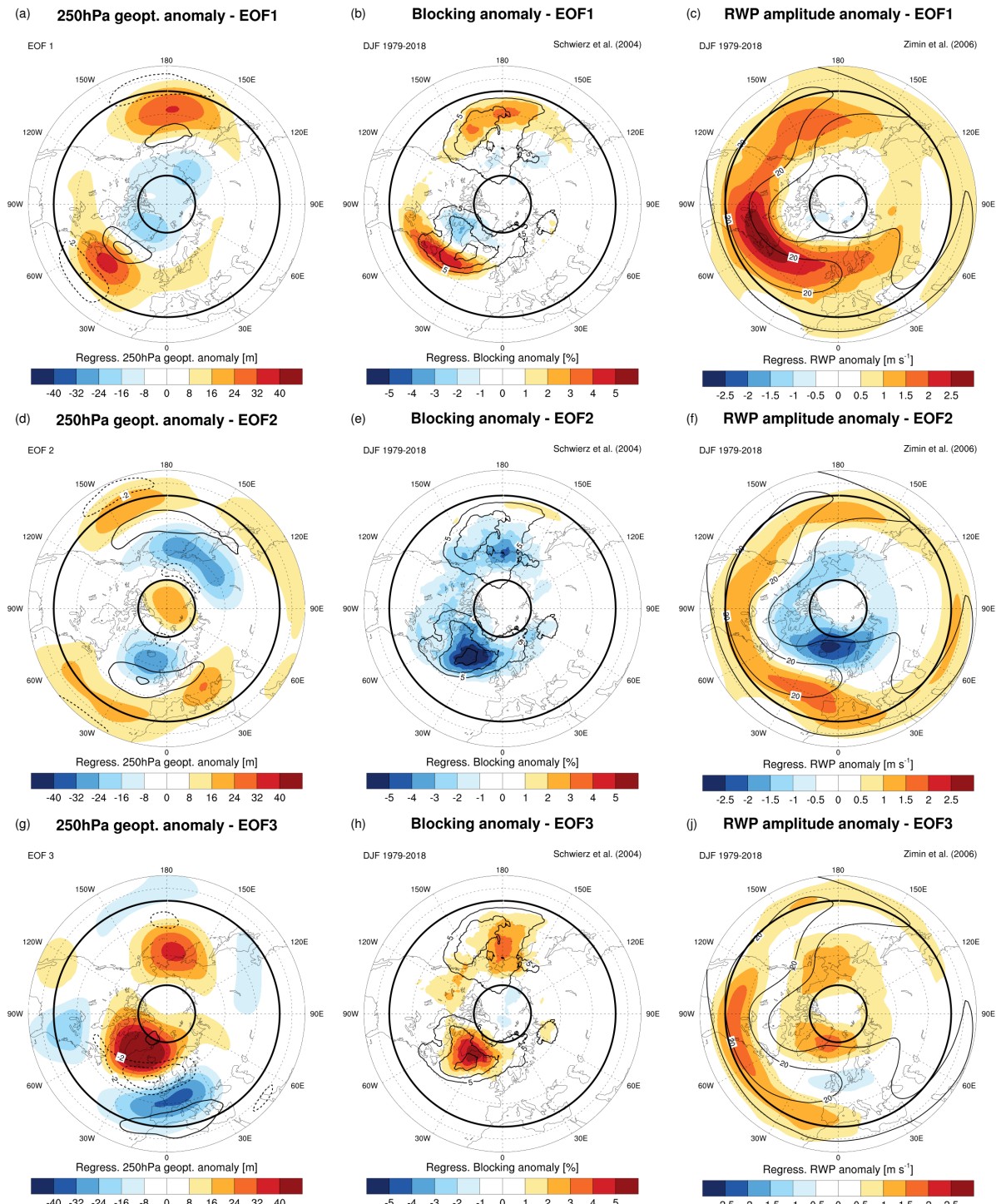

**Figure 3.** Linear regression of anomalies in (a,d,f) 250 hPa geopotential, (b,e,h), blocking frequency and (c,f,j) RWP amplitude with respect to the daily PC time series of (top) EOF1, (middle) EOF2, and (bottom) EOF3. The regressed 250 hPa zonal wind anomalies and the DJF climatologies of blocking and RWP amplitude are overlaid over the respective plots (black contours). Bold black latitude circles indicate 35°N and 75°N.

The remaining part of the paper will then be dedicated to the analysis of the CRWPs associated with events of high PC1 and

280 PC2, which will be denoted as CRWP1 and CRWP2 (events are listed in Table S1, S2 of the Supplement).

Lagged composites of $\mathbf{E}$ and RWP amplitude anomalies are built with respect to the day of maximum PC1/PC2 in each event ($t_{max}$) to visualize the circumglobal character of wave propagation (Fig. 5). Their significance is assessed using a two-sided t-test with respect to the seasonal mean and variance, both extrapolated from daily data. Both CRWPs originate on the Pacific storm track as early as three days before $t_{max}$ (Figs. 5a,e). Significant standardized anomalies of RWP and $E_x$

propagate from the Pacific to the Atlantic at time lags closer to $t_{max}$. Propagation occurs predominantly at high latitudes for PC1 events (Figs. 5b,c), while RWPs stay at lower latitudes during PC2 events (Figs. 5f,g) and eventually enter the subtropical jet over North Africa (Fig. 5h). Interestingly, both PC1 and PC2 events are associated with a stronger than usual upper-level geopotential height gradient across midlatitudes (Figs. 3a,d), a feature that can be interpreted as a proxy for the tropopause-level meridional PV gradient. This is consistent with an enhanced "waveguidability" of the large-scale flow, in the sense of

a configuration that favors the zonal propagation of Rossby waves (Manola et al., 2013; Wirth, 2020) and, therefore, the occurrence of CRWPs.

## 4.1 Origin and evolution

A first question concerns how these CRWPs are generated and in particular which processes, happening in the Tropics and/or in the extratropics, are involved.

**CRWP1.** The analysis of outgoing longwave radiation (OLR) suggests that tropical convection anomalies might play a role in the initiation of CRWP1 events. Lagged composites of pentad-mean OLR indicate the presence of convective activity over the north-eastern Indian Ocean in the 10 days before $t_{max}$ (Fig. 6a). Following the mechanism described by Sardeshmukh and Hoskins (1988), the anomalously strong convection becomes the source of a negative vorticity anomaly at upper levels over the Indian subcontinent that persists while OLR anomalies progressively decay (Fig. 6c). The same anomaly leads to an

acceleration of the subtropical jet stream on its northern flank, as testified by the zonal wind anomalies (Figs. 6a,c). A zoom of Fig. 6 over south-east Asia is portrayed in the supplement (Fig. S3) to better highlight this series of steps. Meanwhile, a large-scale anticyclone intensifies to the east of Japan, together with a cyclonic area in between that anticyclone and the one over the Indian subcontinent, as shown by the geopotential and vorticity anomalies (Fig. 6c). The anticyclone off the coast of Japan is the same one visible in the lagged regression analysis over the North Pacific (Fig. 3a). The arc-shape of the Rossby

wave train originated from tropical convection becomes well visible more downstream, with a trough formed north of Canada and a ridge centred over Newfoundland between lags -2 days and +2 days (Fig. 6e). At positive lags, this Rossby wave train originating from the Tropics disappears and a more zonal propagation of wavenumbers 4 to 5 is noticeable over the midlatitudes (Figs. 6e,g).

The location of the anomalous convection and of the downstream anticyclone are reminiscent of the impact that MJO phase

3 has on the midlatitude circulation (e.g., Jeong et al., 2008; Henderson et al., 2016). To verify possible links, we analyzed the median amplitude and the phase of the MJO during CRWP1 events. The median MJO amplitude was tested for significance using bootstrapping: medians exceeding the 95th percentile of the bootstrapped distribution, obtained after resampling 2500

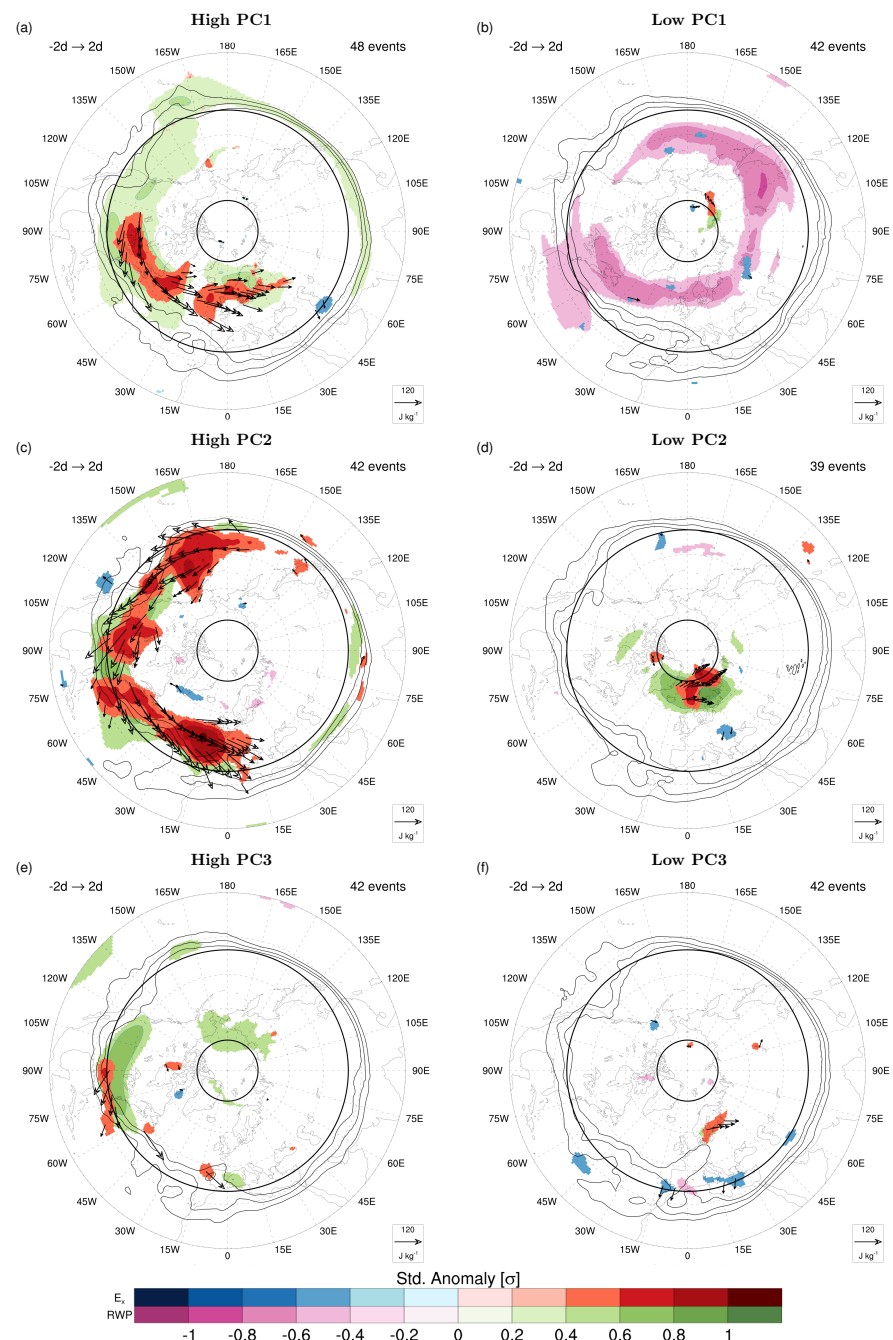

**Figure 4.** Composites of pentad-mean, standardized anomalies of $E_x$ and logarithm of RWP amplitude at $t_{max}$ of events respectively in the top and bottom 15% of (a,b) PC1 (c,d) PC2 and (e,f) PC3. Only significant standardized anomalies ($p < 0.01$) are shown, as well as composite **E** vectors (black arrows) emanating from areas of significant $E_x$ anomalies. Bold black latitude circles indicate 35°N and 75°N. The composite of PV at 250 hPa is overlaid (black contours, only 1 PVU, 1.5 PVU and 2 PVU)

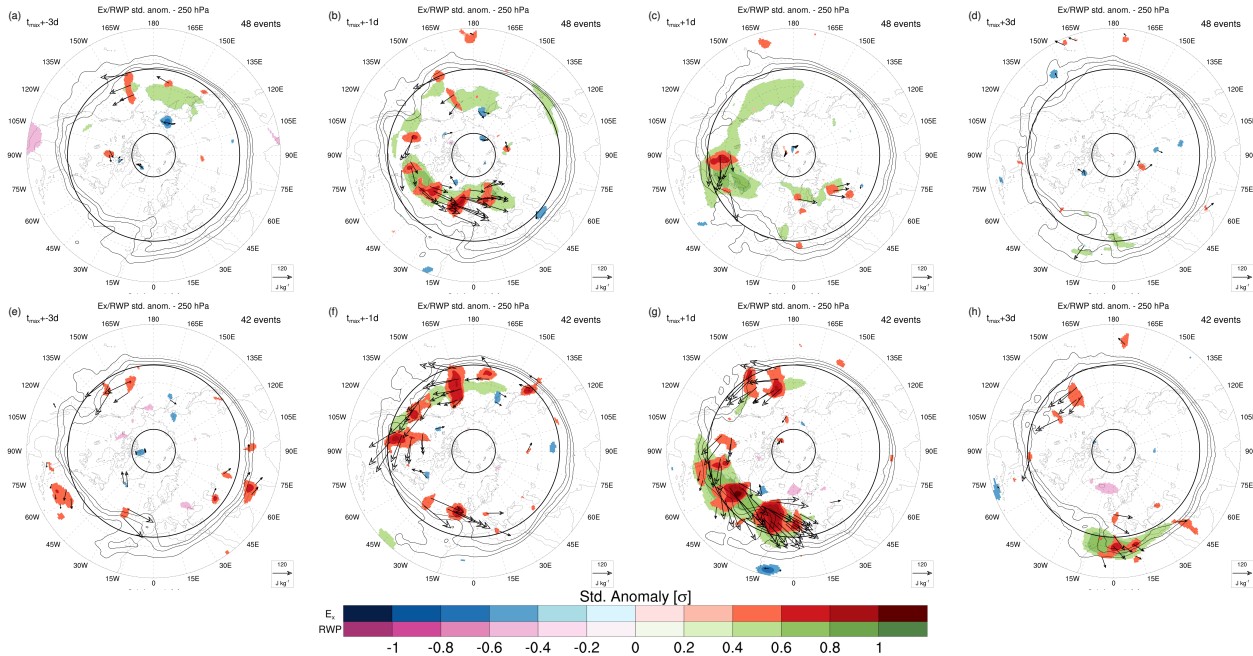

**Figure 5.** As in Fig. 4, but for lagged composites of daily mean, standardized anomalies of $E_x$ and RWP amplitude, for high (top) PC1 and (bottom) PC2 events, at time lags (a,e) $t_{max}$-3 d (b,f) $t_{max}$-1 d (c,g) $t_{max}$+1 d and (d,h) $t_{max}$+3 d.

times, were deemed significant. To take into account possible seasonal variations of MJO amplitude, the calendar days of the random events were selected in a 15-day time window centered on the calendar day of each $t_{max}$, while a random year between 1979 and 2019 (excluding January/February 1979 and December 2019) was attributed. A significantly amplified MJO propagating between phase 3 and 4 is indeed observed 7 days before $t_{max}$ (Fig. 7a). This result is consistent with the lagged composite analysis, as MJO phase 3 corresponds to convection anomalies between India and Indonesia. The dynamical connection between an active MJO phase 3-4 and the activity of the Pacific storm track is achieved by the intensification and extension of the jet stream to the North of the diabatically influenced vorticity anomaly over the Indian subcontinent (Figs. 6a,c). This would induce quasi-geostrophic forcing for ascent and cyclogenesis at the left exit of the jet and, consequently, downstream ridging related to the second anticyclone East of Japan (Jeong et al., 2008). This result is also consistent with previous work that connected phase 3 of the MJO to an enhanced activity of the North Pacific storm track (Guo et al., 2017; Zheng et al., 2018) and to a poleward displacement of the jet stream over midlatitudes, first over the Pacific and then over the Atlantic in the form of a positive phase of the North Atlantic Oscillation (NAO; e.g., Henderson et al., 2016; Lin and Brunet, 2018; Fromang and Rivière, 2020). The positive upper-level zonal wind anomalies visible over central Europe 4 to 8 days after $t_{max}$ (Fig. 6j), however, are not associated with significantly positive NAO at positive lags (not shown).

**CRWP2.** The initiation of CRWP2 does not appear to be clearly related to tropical convection: only weak OLR anomalies are observed over the Indian ocean 8 to 4 days before $t_{max}$ (Fig. 6b), associated with a marginally significant MJO amplitude

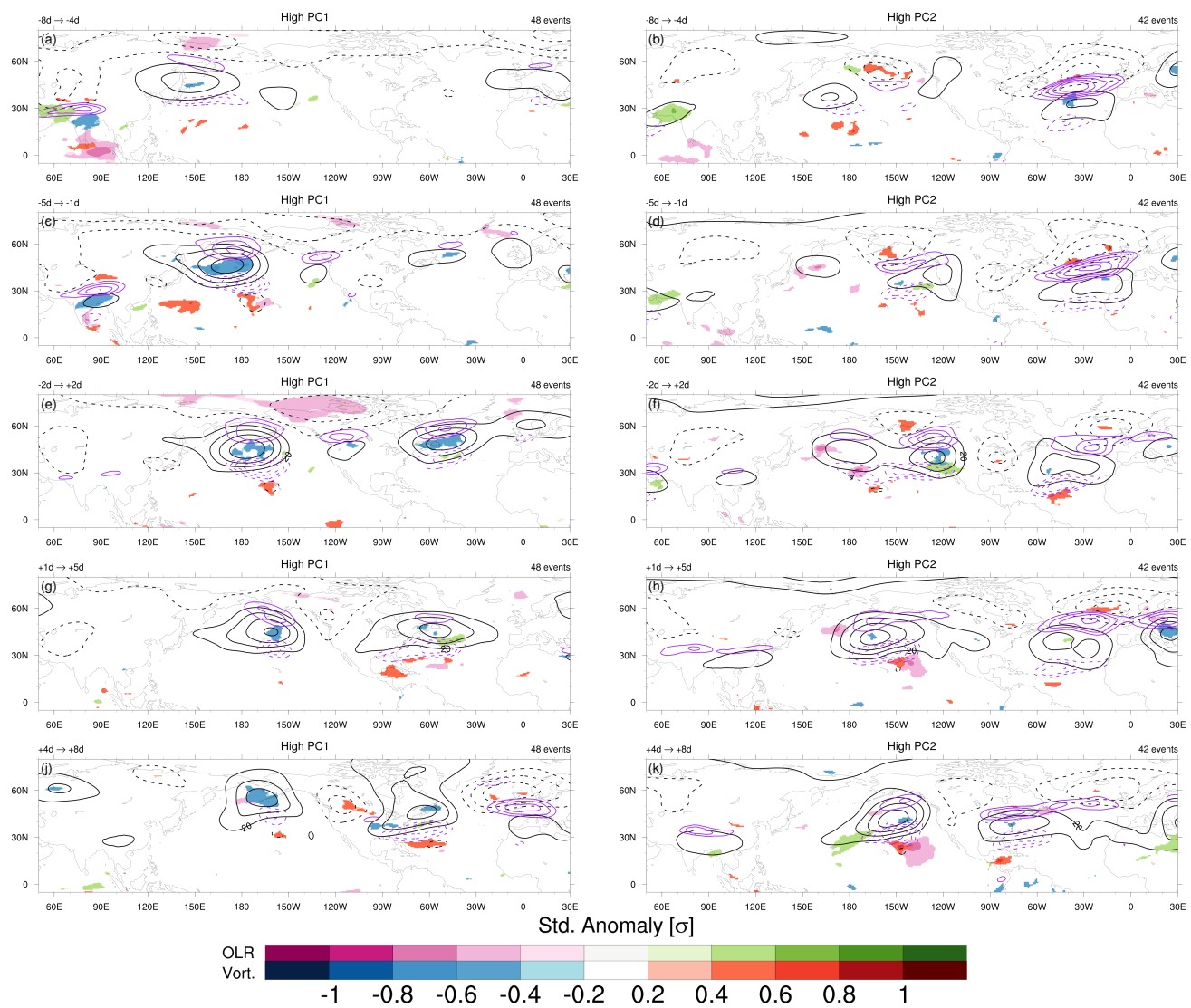

**Figure 6.** Lagged composites of significant ($p < 0.01$, two-sided t-test) pentad-mean standardized anomalies of OLR and 250 hPa relative vorticity (shaded) for (left) CRWP1 and (right) CRWP2 events for the pentads centered at (a,b) $t_{max}$-6 d (c,d) $t_{max}$-3 d, (e,f) $t_{max}$, (g,h) $t_{max}$+3 d and (j,k) $t_{max}$+6 d. Contours of pentad-mean 250 hPa geopotential height anomalies (black contours, between -80 m and +80 m every 20 m excluding zero) and positive 250 hPa zonal wind anomalies (purple contours, between $4\,\mathrm{m\,s^{-1}}$ and $7\,\mathrm{m\,s^{-1}}$ in steps of $1\,\mathrm{m\,s^{-1}}$) are overlaid.

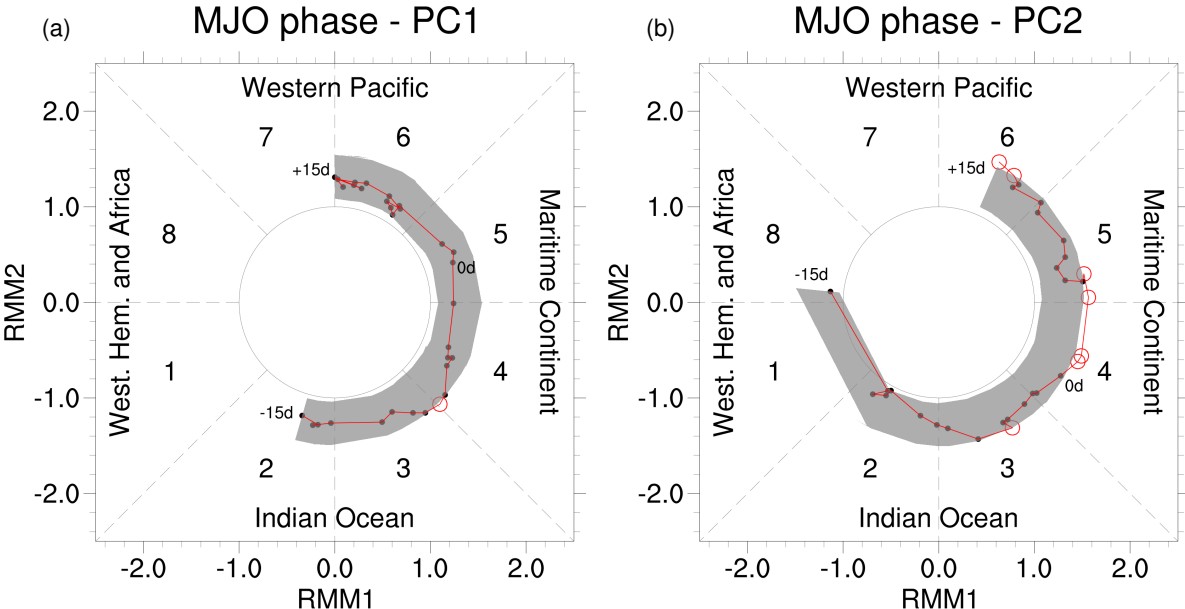

**Figure 7.** Time series of MJO median amplitude, and corresponding phase, for the set of (a) CRWP1 and (b) CRWP2 events, evaluated between $t_{max}$-15 d and $t_{max}$+15 d (connected by red line). The values between the $10^{th}$ and $90^{th}$ percentile of the bootstrapped MJO amplitude distribution are also depicted (grey region): empty red dots indicate amplitudes exceeding the $95^{th}$ percentile of the null distribution.

in phase 3 (Figs. 7b). We notice incidentally that the MJO remains active even after $t_{max}$, potentially indicating that the low-latitude propagation of CRWP2 might affect tropical convection (as suggested already by, e.g., MacRitchie and Roundy, 2016; Chen et al., 2017; Hong et al., 2017). On the other hand, the Icelandic and the Aleutian low are particularly active and located slightly more equatorward than CRWP1 events in the period before $t_{max}$ (Figs. 6d,f). High-latitude positive geopotential anomalies are also visible, as in the regression analysis (Fig. 3d). Although no clear triggering mechanism can be pinpointed for CRWP2, we notice that the equatorward displacement of both the Icelandic and Aleutian Lows leads to a strengthening of the upper-level geopotential gradients at their southern side and to reduction in the tilt of the Atlantic storm track, both factors that could increase the capability to zonally duct RWPs and support the occurrence of CRWPs. The extratropical nature of CRWP2 could also be hinted by the high phase speed of the harmonics involved in it (cf. Fig. 2c), while the tropical forcing leading to CRWP1 would be associated with longer time scales and, therefore, with a projection on slower harmonics.

## 4.2 Effect on meridional heat flux and baroclinicity

Time series of standardized anomalies of 7-day averaged baroclinicity at 775 hPa and transient meridional eddy heat flux at 700 hPa were analyzed to verify the baroclinic nature of wave propagation during CRWP1 and CRWP2 events. These quantities are averaged over three regions: one corresponding to the North Pacific storm track (130°E-180°E, 30°N-50°N), one to the North Atlantic storm track (80°W-30°W, 30°N-50°N, as in Ambaum and Novak, 2014) and one over the 35°N-75°N latitudinal band where spectral analysis was performed (boxes depicted in Fig. 9). As done for MJO amplitude, the median

of the distributions of baroclinicity and heat flux standardized anomalies were tested for significance using bootstrapping. Medians exceeding the 95th percentile of the bootstrapped distribution, obtained after resampling 2500 times, were deemed significant.

**CRWP1.** The passage of CRWP1 is associated with significantly reduced values of baroclinicity over the two main storm track regions: first in the North Pacific in the 8 days preceding $t_{max}$ (Fig. 8a) and then in the North Atlantic in the eight days after $t_{max}$ (Fig. 8c). A significant increase in eddy meridional heat flux occurs in the same time intervals, indicating enhanced poleward heat transport by baroclinic eddies. This increase is particularly strong when considering the broad 35°N-75°N latitudinal belt, although the corresponding decrease in baroclinicity is not statistically significant (Fig. 8b). The dynamical origin of these tendencies is explained by lagged composites of heptad-mean heat flux and baroclinicity. They illustrate how the development of the previously discussed anticyclone over the western North Pacific coincides with significantly enhanced meridional heat fluxes (Fig. 9a). Furthermore, the composites show that the significant reduction of baroclinicity occurs on the southern side of the same anticyclone, as easterlies act to reduce the usual westerly wind shear that is present over the storm track region. A similar mechanism is apparent for the anticyclone subsequently developing over the North Atlantic around $t_{max}$, whose development is preceded by anomalously strong meridional heat fluxes over eastern North America (Fig. 9b) and followed by a significant reduction of baroclinicity at its southern side (Fig. 9c). The full series of lagged composites for CRWP1 has been added to the Supplement (Fig. S4). The amplification of the anticyclones could be due to the enhanced upstream baroclinic activity, indicated by the anomalous heat fluxes, that would result in a meridional amplification of downstream Rossby waves. Given that these anticyclones coincide with low-latitude blocking (Fig. 3b), we also notice that upstream eddy activity is important for blocking development and maintenance, both from a dry dynamics (e.g., Shutts, 1983) and a moist dynamics (e.g., Steinfeld and Pfahl, 2019) point of view. As a final remark, we observe that the spatial pattern of meridional heat fluxes around $t_{max}$ resembles the one regressed onto the Northern Baroclinic Annular Mode (see Thompson and Li, 2015, their Fig. 6b), sparking the hypothesis that the latter can be related to CRWP1.

**CRWP2.** As for CRWP1, CRWP2 events are also associated with anomalously positive peaks in meridional heat flux, first over the North Pacific storm track (Fig. 8d) and then over the North Atlantic (Fig. 8f). These peaks are contextual to a reduction in background baroclinicity, although no significant anomalies are detected. This observation confirms the baroclinic nature of CRWP2, too. Lagged composites depict meridional heat flux anomalies occurring first over the Pacific and then over North America, at latitudes generally lower than PC1 (Fig.,9d-e) and no significant anomalies of baroclinicity in the two storm track regions.

### 4.3 Propagation to the subtropical jet

Lagged composites highlight that enhanced RWP amplitudes persist over North Africa and the middle East at positive lags, while their amplitude keeps decreasing elsewhere (Figs. 5d,h). This would suggest a propagation of RWPs from the midlatitude jet over the Atlantic to the subtropical jet over North Africa. To verify this hypothesis, composites of **E** vector, RWP amplitude and PV are computed for the considered set of events in the days following $t_{max}$.

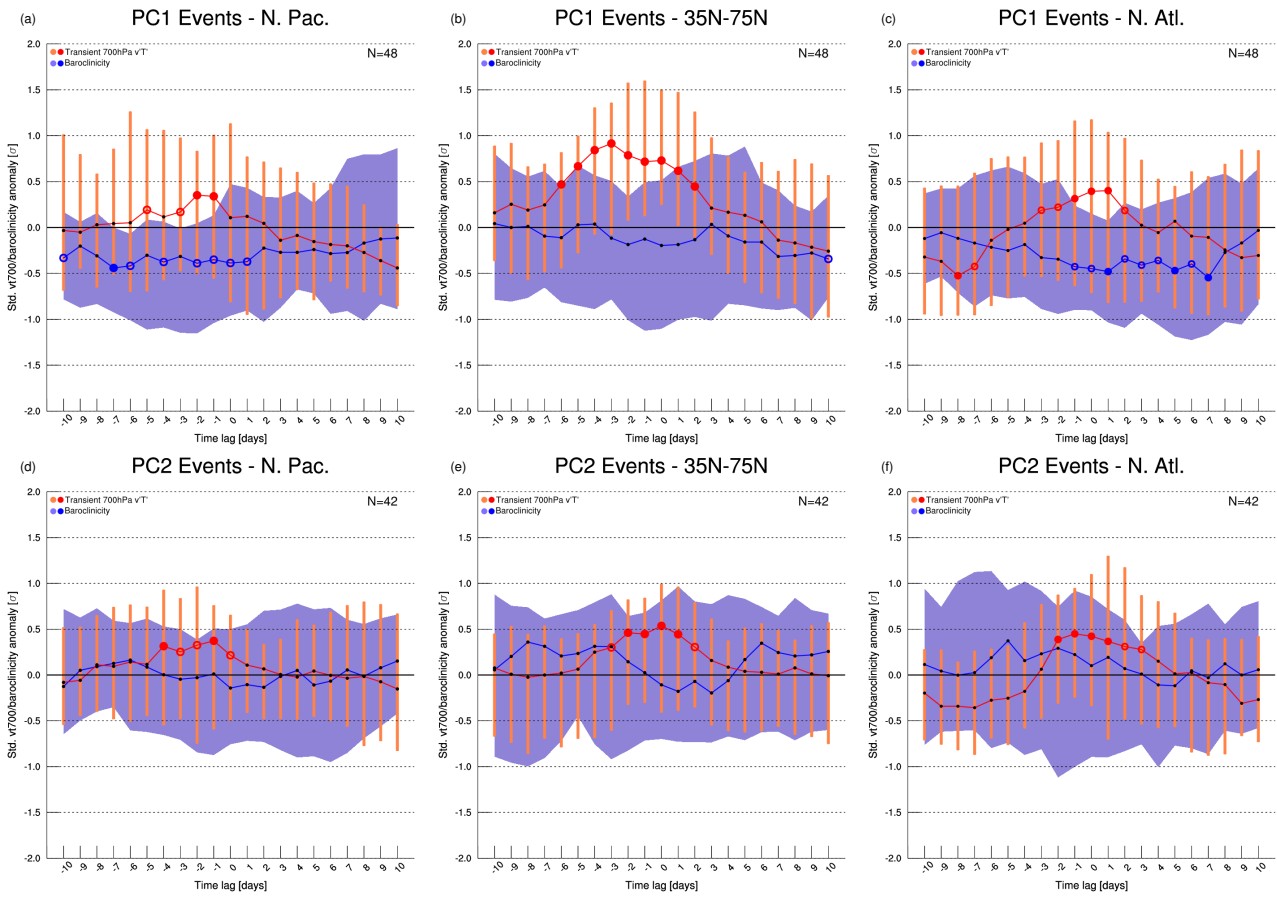

**Figure 8.** Medians of standardized anomalies of 7-day mean, area-averaged baroclinicity $s$ (blue line) and meridional heat flux $v^*T^*$ (red line) for the considered set of (top) CRWP1 and (bottom) CRWP2 events, at different lead times with respect to $t_{max}$, for (a,d) the North Pacific storm track ($130°$E-$180°$E, $30°$N-$50°$N), (b,e) the whole mid-to-high latitudes ($35°$N-$75°$N) and (c,f) the North Atlantic storm track ($80°$W-$30°$W, $30°$N-$50°$N) regions. The length of the orange bars indicates the interquartile range of the heat flux distribution, as well as the width of the blue shading for $s$. Empty (filled) dots indicate anomalies exceeding the $95^{th}$ ($99^{th}$) percentile of the respective bootstrapped distributions.

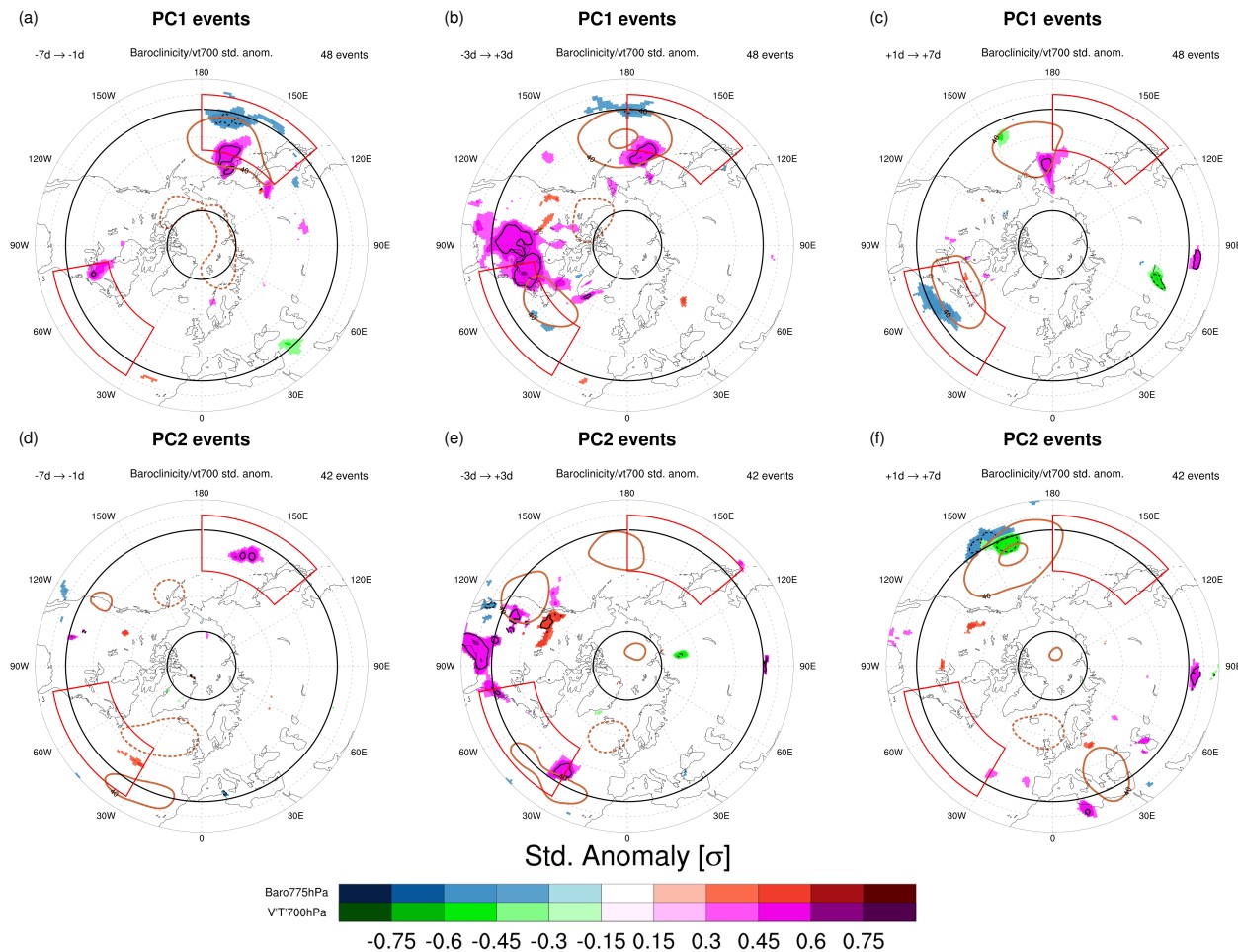

**Figure 9.** Lagged composites of significant ($p < 0.01$, two-sided t-test) heptad-mean standardized anomalies of 775 hPa baroclinicity $s$ and 700 hPa transient meridional heat flux $v^*T^*$ (shaded), together with heptad-mean geopotential height anomalies (brown contours, only -40 m, +40 m and +80 m), for (top) CRWP1 and (bottom) CRWP2 events. Standardized anomalies of $s$ and $v^*T^*$ higher (lower) than $+0.5\sigma$ ($-0.5\sigma$) are contoured by a bold continuous (dotted) black contour. The heptads centered at (a,d) $t_{max}$-4 d (b,e) $t_{max}$ and (c,f) $t_{max}$+4 d are shown. The two storm track regions, where averaging for Fig. 8 was performed, are indicated in red. Bold black latitude circles indicate 35°N and 75°N.

The propagation of CRWP2 and, although to a minor extent, of CRWP1 is associated with significant negative anomalies of the meridional component of **E**, indicating positive momentum fluxes and enhanced anticyclonic orientation of transient eddies at the entrance of the subtropical jet over the Iberian Peninsula and western North Africa (Fig. 10). Following the close link between the group velocity direction relative to the mean flow and the E-vector (Hoskins et al., 1983), the significant anomalies of $E_y$ logically co-occur with the equatorward propagation of significant RWP amplitude anomalies from the North Atlantic to North Africa as seen in the lagged composites of $E_y$ between $t_{max}$+1d and $t_{max}$+3d (Figs. 10a-f). The transfer of RWPs from the North Atlantic to the subtropical jet bears some resemblance with one of the case studies discussed by Ahmadi-Givi et al. (2014), underlining the role played by anticyclonic wave breaking to "close the circle" and connect the Atlantic with the Pacific storm track via the subtropical jet. This result is in agreement with Branstator (2002), who identified a circumglobal teleconnection from a PC analysis over a low-latitude region occupied by the subtropical jet (0-45°N,0-120°E), apparently unrelated to the main storm tracks. Similarly, Davies (2015) was able to close their "weather chain" by tracking wind anomalies at low latitudes along the subtropical jet back to the Pacific storm track, imposing to their pattern a periodicity of approximately 10 days.

## 5   Discussion

### 5.1   Broader implications

The performed analysis indicates that eastward-propagating RWPs are present during periods of CRWP1 and CRWP2. Those RWPs propagate from the Pacific to the Atlantic storm track and then to the subtropical jet over North Africa. Their passage is associated with enhanced meridional heat fluxes, indicative of their baroclinic nature. The wavenumbers involved in the identified CRWPs are consistent with the ones discussed by previous research ($n$=4-6  Branstator, 2002; Davies, 2015) and the large-scale flow configuration features enhanced meridional gradients of geopotential height over both storm tracks, suggesting the presence of jet-level waveguides over a broad portion of the hemisphere (Martius et al., 2010; Manola et al., 2013; Wirth, 2020).

These observations indicate that CRWPs are systematically connected with the propagation of RWPs, at least during boreal winter. More in general, the results support the hypothesis that a CRWP can be shaped by the circumglobal extension of a RWP under large-scale flow conditions featuring a significant "waveguidability" in the zonal direction. In this context, planetary or quasi-stationary waves would still be responsible to shape the hemispheric-scale waveguides and create the conditions favorable to circumglobal RWP propagation (as hinted by Wirth and Polster, 2021). The correctness of this hypothesis, however, would need to be assessed in future research work.

### 5.2   Limitations

A limitation of the employed approach is the fact that harmonics with $c_p$=$0\,\mathrm{m\,s}^{-1}$ are not resolved by the spectral analysis, and quasi-stationary waves are only partly resolved for small zonal wavenumbers (see Supplementary Text S2 for further details).

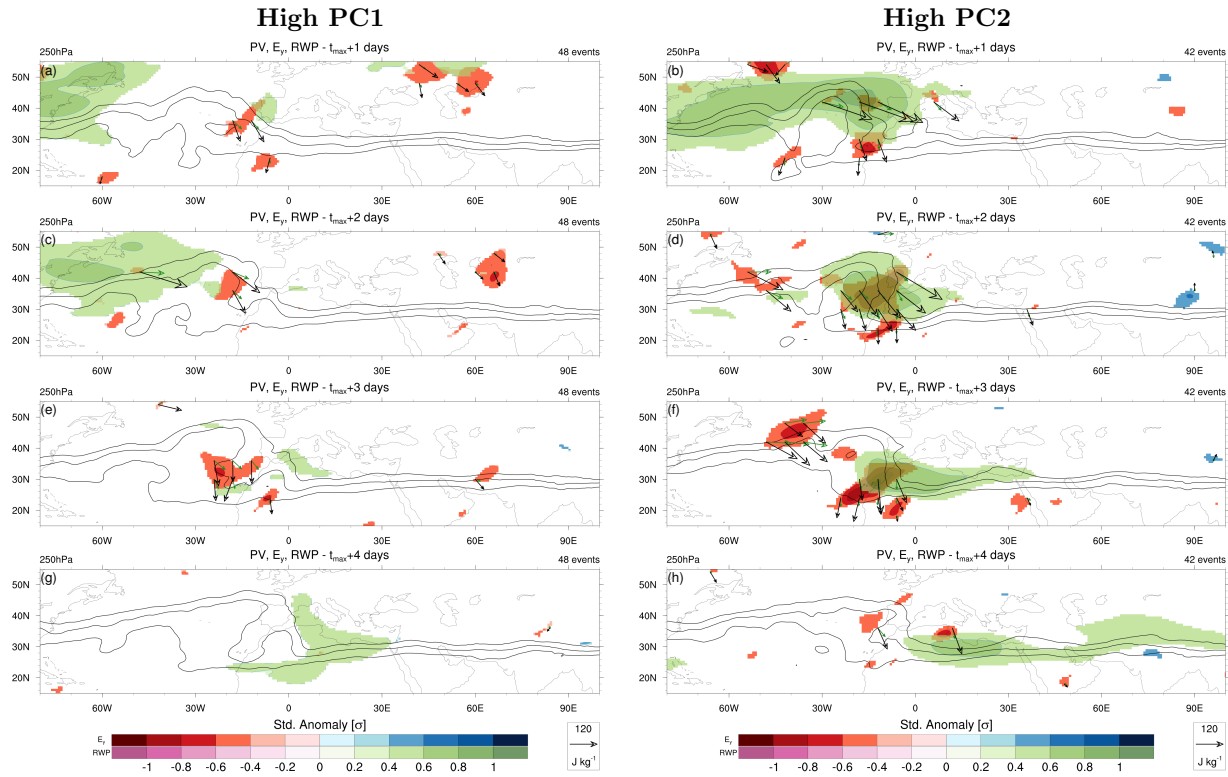

**Figure 10.** Lagged composites of significant ($p < 0.01$) 250 hPa standardized anomalies in $E_y$ (red for anticyclonic tilt, blue for cyclonic tilt of transient eddies) and logarithm of RWP amplitude (green for positive, magenta for negative values) for (left) CRWP1 and (right) CRWP2 events, at (a,b) +1 days, (c,d) +2 days, (e,f) +3 days, (g,h) +4 days with respect to $t_{max}$. The composite of PV at 250 hPa is overlaid (black contours, only 1 PVU, 1.5 PVU and 2 PVU), as well as the composite **E** vectors (black arrows) and DJF mean **E** vectors (green arrows) emanating from areas of significant $E_y$ anomalies.

This limitation could emphasize the role of non-stationary transients with respect to quasi-stationary waves, with the former being picked up more clearly by the EOF analysis. This problematic, however, affects most significantly lower wavenumbers than the ones involved in the identified CRWPs. We cannot fully exclude the existence of a CRWP projecting for most of its lifetime over harmonics with $c_p = 0\,\mathrm{m\,s^{-1}}$ and, therefore, not captured by the employed spectral analysis. These CRWPs could not emerge due to the previously discussed incapability of the space/time spectral analysis to fully resolve quasi-stationary waves. However, the hypothesis that a CRWP described by a perfectly stationary circumglobal wave with wavenumbers $n$=4-6, as for CRWP1, over a period of 61 d (or even of 37 d excluding tapering) seems unrealistic: such a CRWP should feature some degree of propagation, albeit slow, that would be captured by the employed methodology (otherwise it would need to emerge everywhere at the same time). The alternative nature of a quasi-stationary CRWP with respect to a RWP propagating along the jet stream would also need to be theoretically justified: a possibility would be the quasi-resonant amplification of

planetary waves (Petoukhov et al., 2013), a framework whose applicability has been, however, recently questioned (Wirth, 2020).

This study is not a systematic classification of all possible CRWPs. The use of EOF analysis is indeed not appropriate for that purpose, as other CRWPs are bound to appear in higher-order EOFs. In this study we have limited ourselves to the first two EOFs only, which exhibit the simplest patterns in the spectral space and explain the highest variances. More in general, the approach is based on the implicit assumption that CRWPs, if they exist, must feature consistent spectral properties across a broad portion of the hemisphere, strong enough to be picked up by the spectral analysis. The enhancement of few wavenumber/phase speed harmonics in the same time interval would ensure that a single Rossby wave (or RWP, in case of multiple wavenumbers and frequencies) with those harmonics is the one shaping the CRWP. On the other hand, the enhancement of several harmonics in different portions of the spectrum would be an indication of uncoordinated Rossby wave activity across the hemisphere, i.e., not belonging to a CRWP. This assumption might be simplistic: it cannot be excluded, for instance, that uncoordinated RWPs could project by chance on similar wavenumber/phase speed harmonics or that, conversely, a CRWP undergoing strong meridional amplification could project on a broader range of harmonics than expected. The compositing of several CRWP events mitigates the impact of this limitation on the results, but defining more precise spectral fingerprints or adding further constraints (e.g., in terms of RWP location or magnitude across the hemisphere) would likely be needed before applying this approach to compile a synoptic climatology of CRWPs.

## 6 Conclusions and outlook

Wavenumber/phase speed spectral analysis can provide a compact characterization of the Rossby waves propagating above Northern Hemispheric midlatitudes in a given time interval. It allows to assess which harmonics contribute the most to the evolution of the hemispheric flow pattern, evaluating its shape and zonal propagation at the same time. This approach provides a favorable framework to study CRWPs, as the Rossby wave trains constituting them project clearly on distinct harmonics in the spectral domain. We compiled a climatology of the spectral signatures of midlatitude Rossby waves and let CRWPs emerge from it as variability patterns with a prominent, well-defined signal in the spectral domain. The first two spectral variability patterns identified with this approach are indeed related to the zonal propagation of significantly amplified, transient Rossby waves over a significant portion of the hemisphere. Such CRWPs are embedded in a hemisperic-scale circulation pattern characterized by enhanced upper-tropospheric, meridional gradients of geopotential height. We reiterate that these CRWPs emerge from the analysis of spectral properties of midlatitude Rossby waves, without imposing any explicit constraint about the presence of circumglobal waveguides or filtering for specific wavenumbers or time scales.

The first mode of spectral variability during boreal winter corresponds to an eastward-traveling CRWP stretching from the Pacific to the Atlantic storm track and that involves exclusively zonal wavenumbers between $n = 4$ and $n = 6$. The second mode of spectral variability features a selective enhancement of wavenumbers larger or equal than $n = 6$ concomitant to a suppression of slow harmonics with lower wavenumbers, and it corresponds to the rapid eastward propagation of a CRWP over lower midlatitudes. The CRWPs related to these two modes are identified respectively as CRWP1 and CRWP2.

The analyzed CRWPs share some common features and bear some differences. Both originate first in the North Pacific storm track and then propagate to the North Atlantic one. Their propagation is associated with positive anomaly of low-level meridional heat flux over the same regions, which is indicative of the baroclinic activity associated with the traveling Rossby wave packets. Both CRWPs, but in particular CRWP2, propagate from the North Atlantic to the subtropical jet over North Africa, suggesting a potential role of the subtropical jet as a "connector" between the Atlantic and the Pacific storm tracks. Speaking of differences between the two patterns, CRWP2 propagates at overall lower latitudes and its propagation is more rapid than for CRWP1. The main difference, however, lies in their origin: tropical convection in the Indian Ocean likely plays an important role in the initiation of CRWP1, as testified by the significant OLR anomalies registered in the region before CRWP1 events and by the similarity of the circulation pattern to the one following strong MJO events in phase 3. On the other hand, CRWP2 events occur together with a concomitant intensification and slight equatorward shift of the Icelandic and Aleutian low, which might contribute to a reduction in the tilt of the storm tracks (in particular of the Atlantic) and favor a more zonal propagation of RWPs.

We speculate that configurations of enhanced meridional gradients of geopotential height might support the occurrence of CRWPs; this hypothesis needs to be investigated in future work to assess which properties of the large-scale flow can favor circumglobal RWP propagation. A similar analysis to the one performed in this piece of work can be proposed for boreal summer to study warm-season CRWPs: however, performing a circumglobal spectral analysis might not be as appropriate because the two storm tracks are less connected and are associated with weaker waveguides in summer with respect to winter (Branstator and Teng, 2017; Teng and Branstator, 2019). Overall, this study is a first effort to systematically connect midlatitude circulation variability at the hemispheric scale with its spectral signature. This fingerprint of the large-scale circulation can be employed to characterize the shape and propagation of Rossby waves in a global fashion, without the need of filtering, and pave the way to new quantitative tools (as the global phase speed metric developed by Riboldi et al., 2020). Being applied on gridded output of standard meteorological quantities, the method can be easily implemented in climate models to highlight potential biases in the representation of CRWPs or other circulation features, like atmospheric blocking, and assess their possible alterations under different global warming scenarios.

*Author contributions.* JR developed the concept of the project, executed the analysis and wrote the manuscript. All the authors followed the project and supported its technical implementation and the interpretation of the results.

*Competing interests.* The authors declare no competing interests.

*Acknowledgements.* We thank three anonymous reviewers and the editor, Prof. Nili Harnik, for their thoughtful and helpful comments about the first version of the manuscript. The atmospheric blocking data set has been kindly provided by the Institute of Atmospheric and Climate

Science of ETH Zurich. The authors would also like to thank Dim Coumou and Gabriele Messori for the feedback on previous version of this work. This work was supported by funding from the JPI-Climate/Belmont Forum project GOTHAM (ANR-15-JCLI-0004-01 and 01LP1611A).

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
