# Peer review of "Circumglobal Rossby wave patterns during boreal winter highlighted by wavenumber/phase speed spectral analysis"

_Weather and Climate Dynamics, 2021_

## Referee Comment (RC3)

Circumglobal Rossby wave patterns during boreal winter highlighted by wavenumber/phase speed spectral analysis

Authors: Jacopo Riboldi , Efi Rousi , Fabio D'Andrea , Gwendal Rivière , and François Lott

Recommendation: Accept with minor revision

This study systematically identifies two circumglobal Rossby wave packets (CRWPs) using diagnostic analyses with Reanalysis data. The CRWPs were identified with an EOF analysis in wavenumber/phase speed space applied to the daily spectral amplitude of the meridional wind. The authors diagnosed many interesting features of these wave packets, including spatial structure, blocking frequency, and most interestingly a possible driving mechanism via MJO phase 3 tropical convection (for CRWP1) and the generation within the extratropics for CRWP2. The authors also found linkages to the MJO, meridional heat flux, and baroclinicity, connections of the CRWPs to the North Pacific and North Atlantic storm track regions at different time lags, and even to wave packet propagation from the North Atlantic, followed by anticyclonic wave breaking and then propagation across the Mediterranean to the Middle East.  This manuscript was a pleasure to read, and I learned a lot.

I recommend accepting this manuscript with minor revisions and suggest that the authors consider all my comments below.

**Minor Comments**

1. Line 88.  It would be helpful to state here that the seasons will be examined separately, even though it is stated later.
2. Line 112. The spectra are averaged over a very broad range of latitudes. How do the spectra vary if separate averages are performed for more narrow latitudinal bands?  In other words, what would the spectra look like if there were four separate latitudinal bands that are 10 degrees latitudes wide, e.g., 35-45N, 45-55N, etc? Stated slightly differently, how sensitive are the results shown in Fig. 1 to choices of different latitudinal bands.  Note that I am only asking this about Fig. 1.  I don't expect that the authors redo their entire analysis for these four bands, since the results presented in this manuscript are very interesting with this average over more than 40 degrees latitude.
3. Line 113-114.  It is not clear to me how this method gets around the issue of non-zonal propagation. To me, this just a limitation of the method.  Since no method can address all questions, it is sufficient to simply acknowledge this limitation.  Also, another advantage for not first averaging the meridional wind anomalies is that the meridional wind can sometimes be in the opposite direction at different latitudes for the same longitude, which would lead to the cancellation of the signal. Wave breaking and blocks are just two examples when this can happen. The authors may wish to mention this point.
4. Line 117. I don't see how the vertical stacking, i.e., an equivalent barotropic vertical structure, is linked to the need for performing a latitudinal average.  The authors may

wish to explain this more carefully.

5. Line 121.  I suggest that "precise" be replace by "state" or a similar word.

6. Line 136.  It would be clearer to write "The circulation patterns associated with modes of spectral variability…".

7. Figure 1. Westward phase speeds for all wavenumbers less than 8 is a little surprising to me, especially for the larger wavenumbers within this group.  Since the Rossby wave dispersion relation depends on the background zonal-mean zonal wind and beta, or even better, the meridional potential vorticity gradient, are the westward phase speeds related to a small zonal-mean zonal wind within some latitudinal bands (see comment #2 above). Also, does the phase speed in the top left corner of each panel indicate the average phase speed.  This isn't stated in the caption.

8. Lines 153. "Propagation is misspelled.

9. Line 155. "or" -> "of".

10. Lines 190-191.  The authors should provide greater justification for using the meridional component of the E-vector ($E_y$) as a proxy for wave breaking. After all, any horizontal tilt of eddies, no matter how small, will have a non-zero $E_y$.  By looking at observational data, it is very easy to find many days with a fairly large $E_y$ without wave breaking. On the other hand, when there is wave breaking, I would expect $E_y$ to be quite large.  It would be helpful for the authors to show some correspondence between large $E_y$ and wave breaking.

11. Line 226.  Where is the reduced meridional gradient of geopotential that is being referred to?

12. Line 268.  The process described with enhanced tropical convection and a negative vorticity anomaly matches that described by Sardeshmukh and Hoskins (1988, J. Atmos. Sci.) for the so-called Rossby wave source.  It would be good to cite that paper.

13. Line 300. I assume that the authors are referring to Fig. 3d not Fig. 3e.

14. Figure 7.  Since a statistically significant MJO is found at positive lags, and CRWP2 shows anomalies in geopotential height at lower latitudes, it is possible that CRWP2 is exciting the MJO. This isn't surprising since many papers have shown results suggesting that the MJO can be triggered by midlatitude disturbances.

---

## Author Comment (AC2)

**Response to comment by Anonymous Referee #1**

General Comments:

This study explores the boreal winter midlatitude circulation features that emerge from the daily variability modes of the upper-tropospheric zonal wavenumber - phase speed spectrum. The authors focus on the origin and evolution of hemispheric-scale Rossby wave patterns that may characterize the first two modes and test hypotheses regarding the role of tropical convection, jet stream location, and low-level baroclinicity. The analyses are novel and valuable outcomes can be drawn from them. I have a few concerns regarding the basic rationale and scope of this study and some parts of the text require technical clarifications. I recommend a major revision of this manuscript and I hope that the specific remarks listed below will be helpful in this regard.

**Thanks for your comments, which have been very helpful to reflect about the significance of the results and the way they are presented, and for the time spent reading the manuscript. A detailed answer to each comment is given below.**

Specific Comments:

·       The main goal of this study is to investigate the origin and propagation of CRWPs. In some parts of the introduction and the conclusions, it appears that another objective is to showcase the potential of the two-dimensional spectral analysis on the upper-tropospheric circulation. In the end, I am left wondering what is the exact scope of this study and what should be the take-home message for the readers. Perhaps, my confusion lies in the fact that the introduction (lines 14-67) touches on many themes and not all of them are directly relevant to the analyses. Could the authors clarify in the text the research questions adressed in this study and the respective conclusions?

**Thanks for this comment, which allows us to reflect in a deeper way on how the employed methodology serves the scientific objectives of the study. Two aims are intertwined in the paper. The first aim of the study is to highlight how space/time spectral analysis can provide an independent assessment about the existence of CRWPs, without having to explicitly define such circulation patterns in advance and giving insights on their nature. The success of this approach in identifying CRWPs leads to the second aim, that is to study some characteristics of their origin and propagation. The introduction of the paper has been extended and made more precise by explicitly discussing the open research questions in CRWP research. Moreover, possible answers that results of this study can provide to these open questions will be discussed in a new Discussion section.**

·       Following up on the previous comment, if the motivation of the study is to investigate and better understand CRWPs, then I find the employed methodology rather indirect. Instead of constructing daily n-Cp Fourier spectra of the midlatitudes, doing an EOF analysis on these spectra, and focusing on periods when the PC of the leading modes

is large hoping that the composite patterns will exhibit long-lived and/or circumglobal wave patterns, why not directly diagnose CRWP events in the wind field by simply requiring that the RWP amplitude (section 2.2.2) is large enough in a large area for e.g. 5 consecutive days? The second option would result in composite maps that are easier to interpret and actually contain the wave patterns of interest. With the employed methodology, not all the observed CRWP events are necessarily included in the top PCs of EOF1 and EOF2, while these two leading modes may also contain days with no CRWPs, thus "contaminating" the lag-composite analyses with flow configurations beyond the authors' interest.

As correctly noticed, the adopted approach would not necessarily result in the identification of CRWPs. Instead, the principal component analysis is used to highlight the leading modes of variability of meridional wind amplitude in the wavenumber/phase speed spectral space, and it happens that the first two of these modes appear to project onto CRWPs. This result is not surprising, as we expect specific wavenumber/phase speed harmonics to be particularly enhanced during CRWPs (previous work emphasized CRWPs with specific wavenumbers). However, this implies that the discussed CRWPs are likely not the only ones, or that the same CRWP can emerge in different "flavors" in higher EOFs. These limitations and implicit assumptions will be discussed in a new Discussion section in the revised manuscript.

The problem in using the simpler approach proposed by the reviewer is that the occurrence of large RWP amplitudes is not necessarily associated with CRWPs. Large RWP amplitudes could be associated with amplified Rossby waves occurring independently in the two storm tracks, while the basic idea of CRWP would involve a single wave pattern connecting different regions of the hemisphere. Also, we do not expect CRWPs to be necessarily associated with the most extreme RWP amplitudes over the midlatitudes: while there is of course some degree of meridional amplification of troughs and ridges, the really defining aspect of a CRWP is the (significant) zonal extent of its propagation, which is also harder to quantify.

On a more general note, the employed "indirect" approach has this big advantage: it allows to circumvent the challenge of objectively defining CRWPs before their identification. The only implicit assumption is that a CRWP would have a consistent spectral signature evaluated over several latitude circles, meaning that it would project on a defined range of wavenumbers and phase speeds as picked up by the spectral analysis. This allows to identify CRWPs without assuming a-priori how they look like, making the outcome less dependent on subjective choices. That's why we state in the paper that the employed approach lets CRWPs "emerge" from the spectral analysis. Previously employed CRWP diagnostics are affected by the significant difficulties in defining what Rossby waveguides are, difficulties also due to limits in the applicability of standard theoretical frameworks (e.g., the WKB approximation) for waves of significant meridional extent (as reviewed in White et

**al. 2021). The value of the approach adopted here is confirmed by the capability of retrieving large-scale circulation features that other studies discussed in association with the zonal propagation of Rossby waves. A relevant example, discussed in the abstract of the paper too, becomes apparent in the enhanced meridional gradients of geopotential height evident during CRWP1 and CRWP2 events and that likely indicates an enhanced "waveguidability" (as in Wirth et al. 2020). This series of advantages was not mentioned explicitly in the initial version of the paper, and we'll discuss it more comprehensively in the Introduction of the revised version.**

·       Given that the wavenumber-frequency spectra are based on the spectral coefficients (discrete Fourier Transform), writing down their equation would perhaps be more relevant than eq.1 (inverse discrete Fourier Transform). In addition, something seems to be off with the notation in eq.1. Since this is basically a 2D inverse DFT, I would expect the arguments of the spectral coefficient to be [n,j] rather than [n,ω_j] (thus matching the two summation indices), the exponents to be like: exp[i2π(λ*n/N_L + t*j/N_T)], and ω_j to be equal to -2πj/N_T.

**Thanks for this comment, as it gave us the opportunity to improve the clarity and the precision of the Methods section. In general, the choices behind the format and the style of the written formulas reflect the intention to highlight the planetary zonal wavenumber n and the frequency $\omega_j$ , because we use these quantities to define the basis of wavenumber/frequency harmonics described in the methodology.**

**1) We decided not to change the formulation of the inverse Fourier transform to highlight that $\hat{V}'$ are coefficient of the spectral decomposition of the input field $V'$: however, additional informations about what $\hat{V}'$ represent and how they can be used to obtain the spectral power density have been added in the text. 2) The longitude $\lambda$ is actually expressed in radians, so the employed formulation with the planetary, a-dimensional wavenumber n should already be correct: this important information, however, was missing and will be added in the revised version of the paper. 3) Although the index j appears only as a subset $\omega_j$ , we like the employed formulation because it emphasizes that what is in the exponent is actually a frequency, in units of cycles (6h)-1. 4) We would like to keep the minus sign in front of omega in the equation, because in this way the phase speed remains positive when waves are traveling from West to East.**

·       Is the methodology of constructing the wavenumber - phase speed spectra exactly the same as the one presented in Riboldi et al. (2019)? If so, this might be a useful remark for some readers. In any case, some things are not clearly described in the text. I am not sure what is the mathematical formulation behind the phrases: "smoothed 10 times in frequency using a three-point window" (line 103) and "The periodogram is interpolated along lines of constant phase speed" (line 104)? All the results are based on these steps, so it would greatly help the reader if they are described clearly with minimal jargon. ·

Thanks for this comment, the methodology to obtain the spectra involves a series of steps that are, indeed, not always simple to describe. We specified in the text that the procedure followed is exactly the same as in Riboldi et al. (2020), which was however based on meridional wind instead than on meridional wind anomalies. We changed the description of smoothing in frequency using simpler terms to "Each periodogram is smoothed 10 times in frequency with a 1-2-1 running average to further reduce noise". Applications of this very simple filter are found, e.g., in Wheeler and Kiladis (1999) or Small et al. (2014). About the interpolation along lines of constant phase speed, the approach is the one documented by Randel and Held (1991) to transition from wavenumber/frequency spectra to wavenumber/phase speed spectra. A detailed explanation of the interpolation procedure and a schematic of the interpolation have been added to the Supplement of the paper (Text S1, Fig. S1). While working on this point, we noticed that we were occasionally referring to the wavenumber/phase speed periodograms as wavenumber/phase speed spectra. For further clarification, the symbol P has been added to describe periodograms and distinguish them from the spectra (S).

· Why do the days of top 15% PC1/PC2 (that is, up to 30% of all winter days) are included in the CRWP1/CRWP2 lists of events? Doesn't this result in too many events, given that the first 2 EOFs only explain 22.5% of the total variance? In this regard, it is perhaps no surprise that some days are found in both the CRWP1 and CRWP2 events (e.g. 16-24 January 1987, 25-26 December 2014) and that the typical duration of the events (as reported in the Supplement) is much larger that what the evolution in Fig.5 suggests. I presume that taking the top 5% or 10% PC days in each case would produce lists of less events but more indicative(?) of the respective EOF modes. Can the authors elaborate on their rationale behind the choice of this threshold and comment on the sensitivity of the results to its exact value?

The choice of thresholds always features a certain degree of subjectivity: in our case, we chose the upper and lower 15% and a duration of 5 days to have a sufficient balance between intensity of PC events and number of events. The results of the composite analysis, however, are not fundamentally different if the top 10% of PC values, or longer events, are chosen (cf. Fig. R1.1).

[Figure]

*Fig. R1.1: Composites of pentad-mean, standardized anomalies of PV at 250hPa, $E_x$ and logarithm of RWP amplitude at $t_{max}$ of events respectively in the (a) top 20% of PC1 (b) top 10% of PC1 (c) top 20% of PC2 and (d) top 10% of PC2. The number of events is indicated in the top right of each plot. Bold black latitude circles indicate 35°N and 75°N. The composite of PV at 250 hPa is overlaid (black contours, only 1PVU, 1.5PVU and 2PVU). Only significant standardized anomalies (p<0.01) are shown, in Fig. 4 of the manuscript.*

**We will add the following sentence to the discussion of Fig. 4: "The identified patterns do not change substantially if events in the top 20% of top 10% of the corresponding PCs are considered (not shown)."**

It is worth noticing that the decay of composite anomalies after the peak event time is likely due to the compositing procedure and does not exclude that the waves involved in a CRWP can re-circulate a second time around the hemisphere.

About the other point raised in this comment, we think it is not abnormal to have some overlap between events defined starting from the principal component time series. An example could come from the more classic North Atlantic Oscillation (NAO) and East Atlantic (EA) patterns, respectively the first and the second EOF of circulation over the eastern North Atlantic and Europe (Barnston and Livezey 1987). As there is some degree of overlap between the geopotential height/sea-level pressure fields regressed from the two indices, one can envisage that some events of particularly high NAO would also feature high values of the EA index (see, e.g., Mellado-Cano et al. 2019). A similar reasoning could be applied to the patterns regressed from PC1 and PC2, displayed in Fig. After all, the PCs are orthogonal when their correlation is evaluated along the full time series and not necessarily during selected time periods.

· There are some causal statements in the text that - although they appear plausible at first sight - they cannot really be justified by the presented analyses. For example, in lines 267-268 it is stated that, according to Fig.6a, the strong convection is the source of negative vorticity anomaly at upper-levels a few thousand kilometers to the east. Looking at pentad-mean composites doesn't really allow such a statement and the fact that there is a synoptic-scale negative height anomaly in the area (Fig.6a) and at times an active MJO does not correspond to OLR anomalies (line 298) makes me even more hesistant to draw "causal chains" here. Similarly, line 327 reads that the North Atlantic anticyclone development is preceded by anomalously strong meridional heat fluxes over N. America (Fig.9). The problem here is that the significant patterns are rather weak/small and the plotting (contouring choice) may obscure the real sequence of events. Maybe adding the complete series of daily-mean lag composites in the Supplement would shed light on such aspects.

**Role of tropical convection in the genesis of CRWP1 events**

First of all, a zoom on the composite analysis over eastern Asia highlights the negative OLR anomalies to the South-East of India (Fig. R1.2a) nine to five days before $t_{max}$, followed by the generation of a negative vorticity anomaly at 250hPa to the North of it (Fig. R1.2b). As noticed also in a comment by the third referee, the connection between tropical convection and the generation of anticyclonic circulations has been described in the work by Sardeshmukh and Hoskins (1988) about Rossby wave sources, and is consistent with the expected destruction of potential vorticity at upper-levels due to latent heat release.

We propose here two mechanisms to explain the proposed linkage: a first one, based on cyclogenesis theory, and a second one, based on Rossby wave propagation theory.

[Figure]

*Fig. R1.2: Lagged composites of significant (p<0.01, two-sided t-test) pentad-mean standardized anomalies of OLR and 250hPa relative vorticity (shaded) for CRWP1 and events for the pentads centered at (a) $t_{max}$-7d (b) $t_{max}$-5d and (c) $t_{max}$-3d. Contours of pentad-mean 250hPa geopotential height anomalies (black contours, between -80m and +80m every 20m excluding zero) and positive 250hPa zonal wind anomalies (purple contours, 4 m s$^{-1}$ and 6 m s$^{-1}$ isotachs) are overlaid.*

**FIRST MECHANISM: The jet stream is accelerated at the poleward side of the low-vorticity area (Figs. R1.2b,c). As detailed in the manuscript, this would "induce quasi-geostrophic forcing for ascent and for cyclogenesis at the left exit of the jet and, consequently, downstream ridging" which would help inflating the second anticyclone East of Japan: this mechanism was described by Jeong et al. (2008; J. Climate), who evaluated explicitly the forcing for ascent associated with this jet intensification and extension (cf. their Figs. 6, 7). The presence of an anticyclone East of Japan has been reported by other studies that focused on observations and idealized simulations of MJO phase 3 events (e.g., Henderson et al. 2016, cf. their Figs. 3,4, and Fromang and Rivière, cf. their Fig. 2). Unlike other MJO phases, phase 3 is also associated with rainfall over eastern Indian Ocean, very likely of convective nature (cf. the precipitation composites for the different MJO phases produced by the Climate Prediction Center of the NOAA, https://www.cpc.ncep.noaa.gov/products/precip/CWlink/MJO/Composites/Tropical/precip.shtml). Therefore, we believe it is reasonable to assume that an active MJO phase 3 will be portrayed by negative OLR anomalies.**

**SECOND MECHANISM: We evaluated pentad-mean anomalies in wave activity flux (Takaya and Nakamura, 2001) and low-pass geostrophic streamfunction at 250hPa (Fig. R3.1). Following the NCL script by Takaya and Nakamura (available at http://www.atmos.rcast.u-tokyo.ac.jp/nishii/programs/index.html) both these quantities have been computed from daily mean fields by applying a low-pass Lanczos filter with a cutoff of 10 days. The composite over CRWP1 events relative to the pentad between $t_{max}$- 4d and $t_{max}$ indicates anomalously strong wave activity fluxes from the positive streamfunction anomaly over Bay of Bengal towards the anticyclone over the North Pacific, passing through a negative streamfunction anomaly between Korea and Japan. This preliminary analysis is consistent with a role of tropical convection in the initiation of CRWP1 events, although an extratropical contribution (arrows over eastern Russia) appears to be present, too.**

[Figure]

*Fig. R1.3: Composite of pentad-mean low-pass streamfunction anomalies (shaded) and standardized anomalies of wave activity flux with respect to the DJF mean and standard deviation (arrows, plotted only if length is at least 0.5σ) for the pentad centered at t□□□-2d. Data from the ERA5 Reanalysis data set.*

**Anomalously positive transient meridional heat fluxes ahead of upper-level ridges**

**The lagged composites of heat flux and baroclinicity in Fig. 9 were introduced to elucidate the connection between these two quantities from a composite perspective. We believe that the most relevant result, here, is the connection between the baroclinicity reduction and the anticyclones developing next to the entrance regions of the North Pacific and Atlantic storm tracks. These composites provide a different view on the connection between the two quantities described by Ambaum and Novak (2014), as the baroclinicity reduction and the peak in heat fluxes are not co–located but, respectively, at the equatorward and poleward sides of these anticyclones. This arrangement of anomalies persists across the whole period of CRWP1 events (Fig. R1.4, in the next page). The connection between v\*T\* and downstream anticyclones could be due to two processes. 1) Strong baroclinic eddies (as indicated by the anomalous v\*T\*) also result in large-amplitude Rossby waves and the amplification of the anticyclones could be the effect of this enhanced baroclinic activity. 2) These anticyclones coincide with anomalous blocking activity (Fig. 3b): upstream eddies have been proven important for blocking development and maintenance, both from a dry dynamics (e.g., Shutts 1983) and a moist/latent heating dynamics (e.g., Steinfeld et al. 2020) point of view.**

*Fig. R1.4: Lagged composites of significant (p<0.01, two-sided t-test) heptad-mean standardized anomalies of 775 hPa baroclinicity and 700 hPa transient meridional heat flux (shaded), together with heptad-mean geopotential height anomalies (brown contours, only -40m, +40m and +80m), for CRWP1 events. Standardized anomalies higher (lower) than +0.5σ (-0.5σ) are contoured by a bold continuous (dotted) black contour. All the heptads centered between t□□□-6d and t□□□+5d are shown. Averaging regions as in Fig. 9 in the manuscript.*

[Figure]

Std. Anomaly [σ]

Baro775hPa
VT'700hPa

-0.75 -0.6 -0.45 -0.3 -0.15 0.15 0.3 0.45 0.6 0.75

**We made the plots in Fig. 9 easier to interpret by changing the color scale and adding an additional contour line to highlight significant anomalies of at least 0.5 standard deviations.**

Technical Corrections:

·      The abstract ends rather abruptly. It would be worth and nice to add a sentence with the take-home messages of this study. **Thanks for this suggestion, we would add a closing sentence "The obtained results highlight the substantial contribution of propagating Rossby wave packets to CRWPs, hinting that the two features might have the same nature."**

·      Line 14: "Most of the weather systems ... have their dynamical origin ..." **We improved the formulation as suggested.**

·      Line 16: "... form and track across those regions ..." **Thanks, corrected as suggested.**

·      Line 18: "... displacements of the jet ..." **Thanks, corrected as suggested.**

·      Line 20: "... substantially across seasons and years (Grise et al. ..." **Modified to "across seasons and between different years"**

·      Lines 36-37: Is this really "another" circulation pattern? The phenomena mentioned in lines 32-35 may also feature the zonal propagation of RWPs. **Please see the reply to the next comment.**

·      Lines 37-38: The readers should clearly visualize in their minds what exactly are the CRWPs that this study focuses on. The sentence "CRWPs can occur if the large-scale flow configuration is conducive to the zonal propagation of RWPs over long distances" explains when do CRWPs occur; not what they are. So, how do they look like in e.g. a snapshot of the upper-tropospheric wind field? Or is this a concept attached to longer-term means? **This thoughtful comment is particularly appreciated, because previous research did not actually solve this issue, leaving it as an open question. CRWPs are usually visualized in weekly or monthly time means that are sometimes used to build composite means (e.g., Kornhuber et al. 2019). The closest "snapshot" view of a CRWPs is probably the "weather chain" discussed by Davies et al. (2015), which also bears analogies with the so-called recurrent Rossby wave patterns (RRWPs; Röthlisberger et al. 2019, Ali et al. 2021). So, are CRWPs a special type of planetary wave, forming by e.g., quasi-resonant amplification in the presence of especially favorable conditions, or are CRWPs the result of more "known" Rossby wave packets propagating across long distances while maintaining intact their fundamental characteristics (in terms of involved wavenumbers and phase/group speeds)? We will make this issue explicit in the Introduction of the paper.**

·      Lines 58-60: In these lines the benefit of the spectral analysis is briefly mentioned. Can the authors elaborate a bit more on which frameworks make "a-priori assumptions

about the existence of a background flow"? Listing a couple of specific examples upon which the new framework can make a difference would better communicate the significance of this study. **Thanks for this comment, the cited sentence was indeed not clear enough. Instead, we will add a detailed discussion of the advantages that the chosen methodology brings with respect to previous methodologies to study CRWPs. In short, 1) it does not involve any temporal filtering 2) it resolves propagating RWPs as well as slow, planetary waves, although not the stationary ones with $c_p =$ 0m/s and 3) it does not necessitate a-priori definitions of complex dynamical features as circumpolar waveguides. All these advantages allow one to get a complete picture of the waves involved in CRWPs and to discern the role of RWPs in their evolution. More in general, space/time spectral analysis of midlatitude Rossby waves can be adopted to study other categories of Rossby waves that are not circumglobal (e.g., to distinguish in a more objective way planetary from synoptic waves).**

· Line 78: What is meant by "rapid module"? **We changed it to "routine".**

· Line 89: Do the authors mean "... the annual cycle of meridional wind..."? **Thanks for spotting this typo, we corrected it as suggested.**

· Line 105: The phase speed spectra should also be a function of time, right? **Actually the phase speed spectrum is only one for each considered 61-day period, the time evolution of the spectra is implicit in the sense that the wavenumber/phase speed spectrum slowly evolves as the center of the considered time window shifts across each day.**

· Line 110: Which coordinate change is meant here? **We replace this formulation with "transformation from the frequency to the phase speed space", to be more specific while following the terminology by Randel and Held (1991).**

· Lines 119-120: What is meant by "averaging over different latitudes"? The approach where spectra of different latitudes are averaged, or the approach where wind anomalies are averaged over different latitudes? **Thanks for this comment, the averaging refers to the average of spectra across latitudes. We realize that the location of this sentence in the paragraph was probably at the origin of this confusion and that his could be avoided by moving the sentence a few lines above.**

· Line 121: Why is a spectrum attributed to every single day? The temporal resolution is 6-hourly, so I would expect 4 global estimates S(n,cp) per day. Is there a daily averaging step that I missed? **We decided not to compute 4 spectral estimates per day, as the spectra already evolve relatively slowly from one day to another. We will specify in the revised text that only one spectrum per day is computed, centered at 12UTC.**

· Fig.1: Shouldn't the units be [(m/s)^2/Δc]? **Absolutely, thanks for spotting this very important typo.**

· Line 205: The monopole structure in EOF1 (Fig.2a) suggests variability in the amplitude of Rossby waves (with no apparent shift in wavenumber and phase speed) as shown in Fig.4a,b. Is that what the authors also imply by "enhancement of such harmonics"? **The enhancement of harmonics observed in EOF1 does not necessarily imply an increase in the meridional extent of RWPs, even though the latter is observed (as seen in Figs. 4a,b). To avoid misunderstandings, we will refer in the manuscript to "RWP amplitude" only in the sense of meridional amplitude and not in terms of signal amplitude from the spectral perspective. Also, the harmonics with higher than usual power during CRWP1 events span a relatively small range of the climatological spectrum: it is these specific harmonics that are more represented in the flow, rather than the amplitude of Rossby waves in general.**

· Line 214: "... wavenumber co-vary with an ..." **Thanks, rephrased according to the suggestion.**

· Line 257: "Interestingly, both PC1 and PC2 events are ..." **Thanks, rephrased according to the suggestion.**

· Lines 281-283: The procedure to assess the stat. significance of MJO amplitudes is not clear to me. Can the authors reformulate this sentence? **We will rephrase the description as follows, trying to explicit how the bootstrapping can take into account the possible seasonal dependence on MJO amplitude: "The median MJO amplitude was tested for significance using bootstrapping: medians exceeding the 95th percentile of the bootstrapped distribution, obtained after resampling 2500 times, were deemed significant. To take into account possible seasonal variations of MJO amplitude, the calendar days of the random events were selected in a 15-day time window centered on the calendar day of each $t_{max}$, while a random year between 1979 and 2019 (excluding January and February 1979 and December 2019) was attributed."**

· Figs.4,5,6,9,10: The colorbar labels are particularly small and hard to read. I guess a 50% (or more, for Fig.5) increase in the font would suffice. **Thanks for pointing it out, we will increase the labels of the indicated figures to make them more visible.**

· Figs.4,5: Do the black contours depict 250hPa PV as in Fig. 10? **Yes, thanks for noticing the missing field in the caption. Corrected in Fig. 4.**

· Fig.6: In lines 384-385 the authors hypothesize an equatorward shift of the jet for PC2 events. Adding contours for negative zonal wind anomalies might help evaluate this hypothesis. **Thanks for noticing this uncorrect point in the conclusions, as the equatorward shift of the jet was not discussed in the results. We will refer now to a "concomitant intensification and slight equatorward shift of the Icelandic and**

**Aleutian low" as in the rest of the paper. We also followed the suggestion of adding negative zonal wind anomalies to Fig. 6, which has been updated. We will also add a sentence in the results commenting about the possible role of the equatorward shift of the Icelandic low in reducing the tilt of the North Atlantic storm track, thus favoring a more zonal propagation of RWPs.**

·       Lines 308-309: The term "release of baroclinicity" doesn't sound very meaningful. Besides, should the authors briefly describe the Ambaum and Novak (2014) hypothesis, given that big part of this work (section 4.2) is about to test it? Finally, this hypothesis doesn't really talk about a "hemispheric-scale discharge of baroclinicity" as implied in lines 337-338; it focuses on the N. Atlantic storm track. **Thanks for this comment, we decided to not refer anymore to "hemispheric release of baroclinicity" and back away from the hypothesis by Ambaum and Novak (2014), shifting the focus of the section on the assessment of the baroclinic nature of the CRWP.**

·       Lines 310, 315: Why is a 7-day averaging employed here instead of a 5-day one as before? Why is stat. significance assessed based on the 95th percentile instead of the 90th one as before? This inconsistency is somewhat disconcerting and hinders the comparison of Figs. 6 and 9. The maps in these 2 figures could also have the same projection for consistency. **The decision to use 7-day averaging for the baroclinicity stems from the exigence of representing the background baroclinicity in which the transients are developing, which is expressed as a 7-day mean. For consistency, given that we are comparing this metric with the heat flux, we used the same time scale for the heat flux. The polar projection was chosen to facilitate the comparison with Fig. 6b of Thompson and Li (2014), that showed positive anomalies of meridional heat fluxes over similar regions as CRWP1 events during peaks of their Northern Baroclinic Annular Mode (NBAM; cf. Fig. R1.5), and that's why we would prefer to keep it.**

[Figure]

*Fig. R1.5: 1-day lagged regression of eddy fluxes at 850 hPa with respect to the standardized NBAM index. Copy of Fig. 6b from Thompson and Li (2014).*

**In addition, we decided to set the 95th percentile for significance assessment through the whole manuscript for further consistency.**

·       Lines 388-390: Why would there be a problem conducting this spectral analysis in the summer season? Weaker storm tracks may also experience distinct modes of variability. Is there a technical issue I miss? **Thanks for this comment. We will add these citations and clarify better why we expect that performing the same analysis could be more difficult in summer. Branstator and Teng (2017) noticed that waveguidability tends to be confined to separate sectors of the hemisphere during summer. The lower intensity of the jet stream would also reduce its capability to act as a waveguide (Teng and Branstator 2019, Wirth et al. 2020). However, this comment remains a speculation and it could still be that the spectral analysis method would be able to identify CRWPs in summer as well as in winter.**

Bibliography

Ali, S. M., Martius, O., & Röthlisberger, M. (2021). Recurrent Rossby wave packets modulate the persistence of dry and wet spells across the globe. Geophysical Research Letters, 48, e2020GL091452. https://doi.org/10.1029/2020GL091452

Ambaum, M. H. P. and Novak, L.: A nonlinear oscillator describing storm track variability, Quart. J. Roy. Meteor. Soc., 140, 2680–2684, https://doi.org/https://doi.org/10.1002/qj.2352, 2014.

Barnston, Anthony G., and Robert E. Livezey. " Classification, Seasonality and Persistence of Low-Frequency Atmospheric Circulation Patterns", Mon. Wea. Rev. 115 (1987): 1083-1126, https://doi.org/10.1175/1520-0493(1987)115<1083:CSAPOL>2.0.CO;2

Branstator, G. and Teng, H.: Tropospheric Waveguide Teleconnections and Their Seasonality, J. Atmos. Sci., 74, 1513–1532, https://doi.org/10.1175/JAS-D-16-0305.1, 2017

Davies, H.: Weather chains during the 2013/2014 winter and their significance for seasonal prediction, Nature Geosci., 8, 833–837, https://doi.org/10.1038/ngeo2561, 2015.

Fromang, S. and Rivière, G.: The effect of the Madden–Julian Oscillation on the North Atlantic Oscillation using idealized numerical experiments, J. Atmos. Sci., 77, 1613–1635, https://doi.org/10.1175/JAS-D-19-0178.1, 2020

Henderson, S. A., Maloney, E. D., and Barnes, E. A.: The influence of the Madden–Julian Oscillation on Northern Hemisphere winter blocking, J. Climate, 29, 4597–4616, https://doi.org/10.1175/JCLI-D-15-0502.1, 2016.

Jeong, J.-H., Kim, B.-M., Ho, C.-H., and Noh, Y.-H.: Systematic Variation in Wintertime Precipitation in East Asia by MJO-Induced Extratropical Vertical Motion, J.Climate, 21, 788–801, https://doi.org/10.1175/2007JCLI1801.1, 2008

Kornhuber, K., Petoukhov, V., Petri, S., Rahmstorf, S., and Coumou, D.: Evidence for wave resonance as a key mechanism for generating high–amplitude quasi–stationary waves in boreal summer, Clim. Dyn., 49, 1961–1979, https://doi.org/10.1007/s00382-016-3399-6, 2017

Mellado-Cano, Javier, David Barriopedro, Ricardo García-Herrera, Ricardo M. Trigo, and Armand Hernández. " Examining the North Atlantic Oscillation, East Atlantic Pattern, and Jet Variability since 1685", *J. Climate* 32 (2019): 6285-6298, https://doi.org/10.1175/JCLI-D-19-0135.1

Randel, W. J. and Held, I. M.: Phase speed spectra of transient eddy fluxes and critical layer absorption, J. Atmos. Sci., 48, 688–697, https://doi.org/10.1175/1520-0469(1991)048<0688:PSSOTE>2.0.CO;2, 1991.

Riboldi, J., Lott, F., D'Andrea, F., and Rivière, G.: On the Linkage Between Rossby Wave Phase Speed, Atmospheric Blocking, and Arctic Amplification, Geophys. Res. Lett., 47, e2020GL087796, https://doi.org/https://doi.org/10.1029/2020GL087796, 2020.

Röthlisberger, M. Frossard, L., Bosart, L.F., Keyser, D., and Martius, O., " Recurrent Synoptic-Scale Rossby Wave Patterns and Their Effect on the Persistence of Cold and Hot Spells", J. Climate 32 (2019): 3207-3226, https://doi.org/10.1175/JCLI-D-18-0664.1

Sardeshmukh, P. D. and Hoskins, B. J.: The Generation of Global Rotational Flow by Steady Idealized Tropical Divergence, J. Atmos. Sci., 45, 1228–1251, https://doi.org/10.1175/1520-0469(1988)045<1228:TGOGRF>2.0.CO;2, 1988

Small, R., Tomas, R., and Bryan, F.: Storm track response to ocean fronts in a global high-resolution climate model, Clim. Dyn., 43, 805–828, https://doi.org/doi.org/10.1007/s00382-013-1980-9, 2014.

Steinfeld, D., Boettcher, M., Forbes, R., and Pfahl, S.: The sensitivity of atmospheric blocking to upstream latent heating – numerical experiments, Weather Clim. Dynam., 1, 405–426, https://doi.org/10.5194/wcd-1-405-2020, 2020.

Thompson, D. W. J. and Li, Y.: Baroclinic and Barotropic Annular Variability in the Northern Hemisphere, J. Atmos. Sci., 72, 1117–1136, 645 https://doi.org/10.1175/JAS-D-14-0104.1, 2015.

Teng, H. and Branstator, G.: "Amplification of Waveguide Teleconnections in the Boreal Summer", Curr. Clim. Change Rep., 5, 421–432,  https://doi.org/doi.org/10.1007/s40641-019-00150-x, 2019.

Wheeler, M. and Kiladis, G. N.: Convectively coupled equatorial waves: Analysis of clouds and temperature in the wavenumber–frequency domain, J. Atmos. Sci., 56, 374–399, https://doi.org/10.1175/1520-0469(1999)056<0374:CCEWAO>2.0.CO;2, 1999.

White, R. H., Kornhuber, K., Martius, O., and Wirth, V.: From Atmospheric Waves to Heatwaves: A Waveguide Perspective for Understanding and Predicting Concurrent, Persistent and Extreme Extratropical Weather, Bull. Amer. Meteor. Soc., pp. 1–35, https://doi.org/10.1175/BAMS-D-21-0170.1, 2021.

Wirth, V.: Waveguidability of idealized midlatitude jets and the limitations of ray tracing theory, Weather Clim. Dynam., 1, 111–125, https://doi.org/10.5194/wcd-1-111-2020, 2020

---

## Author Comment (AC3)

**Response to comment by Anonymous Referee #2**

This work aims to assess the main observed modes of variability and origins of Rossby wave packets (RWP) in the northern hemisphere winter.

It uses a spectral decomposition method of the upper-level circulation at each latitude to retrieve the most dominant modes of variability in the space of zonal wave number and phase speed. The first two modes are found to be associated with RWPs, and so these are used for further analysis to identify regressions with diagnostics of blocking, large-scale patterns and wave propagation.

The authors also investigate the possible origins of the wavetrains characterizing both modes. They find a likely link between the first mode and tropical convection and the MJO, and elucidate that the second mode is related to extratropical origins, though this link is less clear. They point out other interesting features, such as that both modes exhibit higher synoptic eddy activity and subtropical jet extension that allows the hemispheric wave propagation, which allow a more complete picture of how these modes come about. The paper is clearly written and conclusions well argued. The results contribute significantly to the field of eddy-mean flow interaction and teleconnections in the northern hemisphere. I only have minor comments after which I recommend it for publication.

 **Thanks a lot for your constructive comments and for the time spent reading the paper. A detailed answer to each comment is given below.**

General minor points:

- what is the sensitivity to the windows chosen for the spectral decomposition, and same for the thresholds chosen to isolate PC extremes?

**The choice of the time window is one of the factors determining the type of harmonics that the spectral analysis is capable of resolving. This happens because the length of this time window $T_W$ determines the minimum frequency $\omega_{min} = 2\pi/T_W$ resolved by the spectral analysis and, therefore, the minimal phase speed $c_{min} = \omega_{min}\, acos(\phi)\,/n$ of the wavenumber/phase speed harmonics that can be resolved. In general, the larger $T_W$ the smaller $\omega_{min}$ and $c_{min}$ become: however, increasing too much the size of the time window would mean that several days are analyzed at once, smoothing the day-to-day spectral variability from which CRWPs are emerging. A simpler analogy to understand this problem would be the one of taking averages over blocks of days, for instance to diagnose the occurrence of heatwaves by taking extremes in 2-m temperature over a given region. One can choose to do so for overlapping bi-weekly, monthly or two-monthly time intervals: shorter time windows would emphasize extreme warm spells, while longer time windows would emphasize persistent warm spells that are not necessarily the most intense (the choice of the appropriate time window depends, of course, on the research question that is being addressed).**

Coming back to the specific case addressed in this work, we show here how the seasonal mean (DJF) spectral power density changes as it is computed over time windows with different lengths: 41d, 61d (the one employed in the paper) and 81d (Fig. R2.1). The double cosine tapering is adjusted to always match the first 20% and the final 20% of the days in each time interval.

[Figure]

*Fig. R2.1: DJF mean of wavenumber/phase speed spectra of meridional wind anomaly at 250hPa, obtained from spectral analysis performed over different consecutive time intervals: (a) 41 days, (b) 61 days and (c) 81 days. Units and notation as in Fig. 1 of the manuscript.*

It can be noticed that reducing $T_W$ from 61d to 41d increases the average spectral power density everywhere, in particular for rapid transients with high wavenumbers and phase speeds (Fig. R3.2a). This is due, first of all, to the higher variability of the flow in this shorter time interval and is consistent with Parseval's identity. It is also likely due to a reduced capability of resolving low phase speeds for harmonics with low wavenumbers: this is shown by the broader area of missing values in Fig. R3.2a. Moving to larger $T_W$ reduces this effect: given that the area with unresolved harmonics does not vary between 61d and 81d, we chose 61d to retain some variability while being able to resolve most harmonics at low wavenumbers. The EOF analysis was also repeated after having used a time window of 81d to compute the spectra, in order to better resolve slow-moving harmonics, and the first two EOF patterns remained substantially the same.

The latitude chosen for the spectral analysis also affects the resolved harmonics: a Supplementary Text S2 in Supplementary material will be added to explain these effects in detail (see also the comment raised to the second point of the third reviewer).

About the thresholds chosen to identify CRWP events, their choice always features a certain degree of subjectivity: in our case, we chose the upper and lower 15% and a duration of 5 days to have a sufficient balance between intensity of PC events and number of events. The results of the composite analysis are not fundamentally different if the top 10% of PC values, or longer events, are chosen (cf. Fig. R2.2).

[Figure]

*Fig. R2.2: Composites of pentad-mean, standardized anomalies of PV at 250hPa, $E_x$ and logarithm of RWP amplitude at $t_{□□□}$ of events respectively in the (a) top 20% of PC1 (b) top 10% of PC1 (c) top 20% of PC2 and (d) top 10% of PC2. The number of events is indicated in the top right of each plot. Bold black latitude circles indicate 35°N and 75°N. The composite of PV at 250 hPa is overlaid (black contours, only 1PVU, 1.5PVU and 2PVU). Only significant standardized anomalies (p<0.01) are shown, in Fig. 4 of the manuscript.*

- As I understand it, the authors use the EOFs of the spectral decomposition as the basis for their composites and regressions, because they want to analyse regimes based on the different wavenumbers/phase speeds, groups of which we know have different characteristics (e.g., planetary waves vs synoptioc waves). I like that the analysis is based on reducing dimensionality for physical reasons, but I wonder whether some of the composites, e.g. Fig. 9, would yield a less noisy time series if the v*T* and baroclinicity were directly related to

the RWP? It seems that especially Fig. 8(d-f) yield consistent patterns, but with the spread being so large, the authors conclude that these consistent changes in baroclinicity are not significant. At least comment outlining the disadvantages of this technique would be useful. **Thanks for this comment, we will discuss the limitations of employing EOF analysis in a new Discussion section, added between the Results and the Conclusions sections.**

- the Figure labels could be enlarged. **True, thanks for noticing it. We will increase the size of labels in Figures.**

Specific clarifications

- L20: between one year or the other - you mean inter-annually? **Yes, we reworded it to "across seasons and between different years".**

- The next three points aim to differentiate between the two types of mechanisms more clearly: - L22: single storm track (internal mechanism)…between the two (hemispheric Mechanisms [or something like that]) - L23: AN internal mechanism - L27: [new line] On the other hand… storm tracks (hemispheric mechanism) **Thanks for these comments, we will rewrite the lines specifying the difference between local and hemispheric mechanisms.**

- L89: please state how the smoothing was performed **We employed a 30-day moving average: it will be stated explicitly in the text and rephrased it to "further smoothed using a 30-day moving average".**

- Eq. 5: it may be worth adding a comment that vT is proportional to the vertical E-vector component and will also be investigated. **Even though this is true, we are a bit reluctant to talk about the Eliassen-Palm flux and the vertical propagation of Rossby waves here because these topics are not discussed in the rest of the manuscript. Therefore, for simplicity, we would like to leave the description as it is.**

- L104: "interpolated along lines of constant phase speed" What exactly do you mean? **Thanks for this comment. The approach is the one used in the paper by Randel and Held (1991), the main reference for wavenumber/phase speed spectra. We will add a detailed explanation of the interpolation procedure to the Supplementary Material.**

- L186: it may be worth including Orlanski (1998) as a reference here, for the interpretation of the horizontal E vectors (https://journals.ametsoc.org/view/journals/atsc/55/16/1520-0469_1998_055_2577_pdost_2.0.co_2.xml) **Thanks a lot for this suggestion, the reference will be added.**

- Fig. 4: I couldn't see any green arrows **Thanks for pointing it out, given that they were not actually discussed in the paper we decided to remove them from Figs. 4 and 5.**

---

## Author Comment (AC4)

**Response to comment by Anonymous Referee #3**

Circumglobal Rossby wave patterns during boreal winter highlighted by wavenumber/phase speed spectral analysis

Authors: Jacopo Riboldi , Efi Rousi , Fabio D'Andrea , Gwendal Rivière , and François Lott

Recommendation: Accept with minor revision

This study systematically identifies two circumglobal Rossby wave packets (CRWPs) using diagnostic analyses with Reanalysis data. The CRWPs were identified with an EOF analysis in wavenumber/phase speed space applied to the daily spectral amplitude of the meridional wind. The authors diagnosed many interesting features of these wave packets, including spatial structure, blocking frequency, and most interestingly a possible driving mechanism via MJO phase 3 tropical convection (for CRWP1) and the generation within the extratropics for CRWP2.

The authors also found linkages to the MJO, meridional heat flux, and baroclinicity, connections of the CRWPs to the North Pacific and North Atlantic storm track regions at different time lags, and even to wave packet propagation from the North Atlantic, followed by anticyclonic wave breaking and then propagation across the Mediterranean to the Middle East. This manuscript was a pleasure to read, and I learned a lot.

I recommend accepting this manuscript with minor revisions and suggest that the authors consider all my comments below.

**Thanks a lot for your positive and constructive comments and for the time spent reading the manuscript. A detailed answer to each point is enclosed below.**

Minor Comments

1. Line 88. It would be helpful to state here that the seasons will be examined separately, even though it is stated later. **The paper focuses on boreal winter, as indicated in the title, but the spectral analysis has been actually done for each day of each season between March 1979 and November 2019. Days belonging to neighboring months (e.g., February 1979 and December 2019) were always considered in the 61d time window needed to perform the spectral analysis, so the computation of the spectra is seamless across the Reanalysis period. Seasons are portrayed separately for Fig. 1 only, to illustrate the climatology. We will specify this aspect for further clarity.**

2. Line 112. The spectra are averaged over a very broad range of latitudes. How do the spectra vary if separate averages are performed for more narrow latitudinal bands? In other words, what would the spectra look like if there were four separate latitudinal bands that are 10 degrees latitudes wide, e.g., 35-45N, 45-55N, etc? Stated slightly differently, how sensitive are the results shown in Fig. 1 to choices of different latitudinal bands. Note that I am only asking this about Fig. 1. I don't expect that the authors redo their entire analysis for

these four bands, since the results presented in this manuscript are very interesting with this average over more than 40 degrees latitude.

**This question is interesting under several aspects, because the choice of the latitudinal band is a factor determining the type of harmonics that the spectral analysis is capable of resolving. This is visible in the $cos(\phi)$ dependence of the phase speed:**

$$c_p = \omega \, acos(\phi) \,/n$$

**and means that the same angular frequency $\omega$ would correspond to a higher phase speed at low latitudes and a lower phase speed at high latitudes. This effect was already discussed by Randel and Held (1991) and leads to the exclusion of few wavenumber/phase speed harmonics from the spectrum. A Supplementary Text S2 has been added to the Supplementary material to explain why this is the case and explicit the resolved range of phase speed.**

**In order to highlight this effect, we show here the seasonal mean (DJF) of spectral power density computed for different latitudinal ranges (Fig. R3.1). The plot to the left of the panel (Fig. R3.1a) is obtained by averaging only the latitudes between 35°N and 55°N, the plot on the right (Fig. R3.1c) by averaging between 55°N and 75°N. For comparison, the plot in the middle (Fig. R3.1b) is the same as in Fig. 1a, obtained by averaging across the full latitudinal range (35°N-75°N).**

[Figure]

*Fig. R3.1: DJF mean of wavenumber/phase speed spectra of meridional wind anomaly at 250hPa, obtained by averaging together periodograms computed across different latitude ranges: (a) 35°N-55°N, (b) 35°N-75°N, (c) 55°N-75°N. Units and notation as in Fig. 1 of the manuscript.*

**We can notice that low-latitude spectra do not resolve properly slow-moving waves for low wavenumbers, as indicated by the relatively large area of missing values around the $c_p = 0$ m/s line (as discussed in Supplementary Text S2). They also project over higher wavenumbers at low latitudes than at high latitudes: this is due to the horizontal scale of the anomalies (which is determined by physical processes like baroclinic instability) projecting on different zonal wavenumbers according to the latitude circle over which the spectral decomposition is performed. For instance, the latitude circle at $\phi_1$= 36°N has a**

length $L_1 = acos(36°) \approx 5000$ **km, while the length at** $\phi_2 = 72°$**N is** $L_2 = acos(72°) \approx 2000$ **km: a ridge/trough couplet with a horizontal scale of 1000 km would then project on a wavenumber n=5 at** $\phi_1$ **and on a zonal wavenumber n=2 at** $\phi_2$ **. These artifacts are an unavoidable limitation of performing spectral analysis over a latitude circle on a sphere: the averaging of the periodograms over the broad 35°N-75°N latitude range reduces their impact on the results. This beneficial effect can be noticed by looking at Fig. R3.2, which features a reasonable distribution of spectral power over the whole range of expected harmonics. We will highlight this effect in the text as an advantage of latitudinal integration, instead of the previously discussed "non-zonal propagation".**

3. Line 113-114. It is not clear to me how this method gets around the issue of non-zonal propagation. To me, this just a limitation of the method. Since no method can address all questions, it is sufficient to simply acknowledge this limitation. Also, another advantage for not first averaging the meridional wind anomalies is that the meridional wind can sometimes be in the opposite direction at different latitudes for the same longitude, which would lead to the cancellation of the signal. Wave breaking and blocks are just two examples when this can happen. The authors may wish to mention this point. **Thanks for pointing it out, we will remove this reference to the non-zonal propagation. For the second comment, please see the next point.**

4. Line 117. I don't see how the vertical stacking, i.e., an equivalent barotropic vertical structure, is linked to the need for performing a latitudinal average. The authors may wish to explain this more carefully. **Thanks for pointing out this unclarity: we used the verb "stacked" inappropriately to indicate a meridional superposition of the anomalies, rather than a vertical superposition. This corresponds to what the reviewer was indicating in comment #3, as meridional wind anomalies can have opposite signs at the same longitude. We will employ the suggested formulation in the manuscript and substitute "stacking" with "meridional superposition".**

5. Line 121. I suggest that "precise" be replace by "state" or a similar word. **Thanks, replaced according to the suggestion.**

6. Line 136. It would be clearer to write "The circulation patterns associated with modes of spectral variability…". **Thanks, rephrased according to the suggestion.**

7. Figure 1. Westward phase speeds for all wavenumbers less than 8 is a little surprising to me, especially for the larger wavenumbers within this group. Since the Rossby wave dispersion relation depends on the background zonal-mean zonal wind and beta, or even better, the meridional potential vorticity gradient, are the westward phase speeds related to a small zonal-mean zonal wind within some latitudinal bands (see comment #2 above). Also, does the phase speed in the top left corner of each panel indicate the average phase speed. This isn't stated in the caption. **This is another interesting observation: we guess that these westward-propagating harmonics occur in correspondence of Rossby wave breaking, when troughs and ridges at the scale of a Rossby wave packet start to move from the east to the west following the nonlinear "bending" of the waveguide. Another**

possible explanation might involve the occurrence of atmospheric blocking, which often appears as a large-scale anticyclonic anomaly that is not part of a normal Rossby wave packet. In this case, the spectral analysis would project power over a broad range of wavenumbers (as the extreme case of a Fourier transform of Dirac's delta would project power on all wavenumbers). JR undertook a preliminary analysis to study the spectral signature (in terms of wavenumber/phase speed harmonics) of atmospheric blocking events and noticed that such events (defined with respect to the blocking index by Schwierz et al. 2004) feature anomalous spectral power for quasi-stationary and retrogressive harmonics (Fig. R3.2).

[Figure]

*Fig. R3.2: Anomalous spectral density for N=351 days featuring the highest area (top 10%) covered by atmospheric blocking in the Northern Hemisphere (anomalies computed with respect to the seasonal cycle). This figure belongs to a poster that was discussed in the Blocking Workshop 2021: the poster is publicly available at the web page https://jriboldi.github.io/files/BW2021_Poster_Riboldietal.pdf or upon request to the corresponding author.*

We will add a sentence to the manuscript to explicit this point: "We speculate that the power in those retrogressive harmonics is due to the occurrence of atmospheric blocking and Rossby wave breaking, phenomena that feature a disruption of the normal eastward propagation of upper-level troughs and ridges".

The [c] value indicates the value of phase speed (computed as in Riboldi et al. (2020) extrapolated from the climatological spectrum. Both [c] and [n] are obtained as weighted averages of phase speed and wavenumber with respect to the spectral power:

**this will be made explicit in the caption by the new formulation "Both these quantities are obtained by weighting, respectively, the phase speed and the wavenumbers in the considered ranges with respect to the spectral power, and the respective values are reported in the top left corner of each plot."**

8. Lines 153. "Propagation is misspelled. **Thanks, corrected.**

9. Line 155. "or" -> "of". **Thanks, corrected.**

10. Lines 190-191. The authors should provide greater justification for using the meridional component of the E-vector (Ey) as a proxy for wave breaking. After all, any horizontal tilt of eddies, no matter how small, will have a non-zero Ey. By looking at observational data, it is very easy to find many days with a fairly large Ey without wave breaking. On the other hand, when there is wave breaking, I would expect Ey to be quite large. It would be helpful for the authors to show some correspondence between large Ey and wave breaking. **Although we are aware that significant anomalies Ey do not necessarily imply wave breaking, we chose Ey for a few reasons: 1) it is easier to compute and based on fewer assumptions than an actual wave breaking diagnostic 2) it matches the employment of the zonal component of the E-vector (Ex) to highlight CRWPs 3) it is not a 0/1 field as the blocking, making it easier to build composites and test its significance. The usefulness of this quantity to highlight eddy/mean flow interaction was already shown by previous work (e.g., Schemm et al. 2018). To avoid misunderstandings with actual wave breaking, we will refer to "anomalous equatorward propagation of transient eddies". The second-last sentence of the abstract will be modified as follows "An anomalous equatorward propagation of Rossby waves from the Atlantic eddy-driven jet to the North African subtropical jet is observed for both CRWPs". Further references to anticyclonic wave breaking will be removed from the text.**

11. Line 226. Where is the reduced meridional gradient of geopotential that is being referred to? **We noticed in Fig.3 an increased meridional gradient of geopotential height, also reflected in the positive upper-level zonal wind anomalies at 250hPa. We will further explain the connection between the two in the text.**

12. Line 268. The process described with enhanced tropical convection and a negative vorticity anomaly matches that described by Sardeshmukh and Hoskins (1988, J. Atmos. Sci.) for the so-called Rossby wave source. It would be good to cite that paper. **We will add the reference to the suggested position, thanks for pointing it out: it indeed is the basis to describe the physical mechanism involved.**

13. Line 300. I assume that the authors are referring to Fig. 3d not Fig. 3e. **Yes, thanks a lot for spotting this typo and allowing us to correct it swiftly.**

14. Figure 7. Since a statistically significant MJO is found at positive lags, and CRWP2 shows anomalies in geopotential height at lower latitudes, it is possible that CRWP2 is exciting the MJO. This isn't surprising since many papers have shown results suggesting that

the MJO can be triggered by midlatitude disturbances. **Thanks for pointing this hypothesis out, we will add it to the text together with a few references to support it.**

Bibliography

Randel, W. J. and Held, I. M.: Phase speed spectra of transient eddy fluxes and critical layer absorption, J. Atmos. Sci., 48, 688–697, https://doi.org/10.1175/1520-0469(1991)048<0688:PSSOTE>2.0.CO;2, 1991.

Riboldi, J., Lott, F., D'Andrea, F., and Rivière, G.: On the Linkage Between Rossby Wave Phase Speed, Atmospheric Blocking, and Arctic Amplification, Geophys. Res. Lett., 47, e2020GL087796, https://doi.org/https://doi.org/10.1029/2020GL087796, 2020.

Schemm, S., Rivière, G., Ciasto, L. M., and Li, C.: Extratropical cyclogenesis changes in connection with tropospheric ENSO teleconnections to the North Atlantic: Role of stationary and transient Waves, J. Atmos. Sci., 75, 3943–3964, https://doi.org/10.1175/JAS-D-17-0340.1, 2018.

Schwierz, C., Croci-Maspoli, M., and Davies, H. C.: Perspicacious indicators of atmospheric blocking, Geophys. Res. Lett., 31, https://doi.org/10.1029/2003GL019341, l06125, 2004.

---

## Author Response (AR1)

**Circumglobal Rossby wave patterns during boreal winter highlighted by wavenumber/phase speed spectral analysis**

Jacopo Riboldi, Efi Rousi, Fabio D'Andrea, Gwendal Rivière, and François Lott

**Submitted to Weather and Climate Dynamics**

**Author's response**

We would like to thank the three anonymous reviewers and the editor for their thorough and helpful comments on the preprint of the manuscript. The main conclusions remain fundamentally unchanged, but we feel that the scope of the paper has been made clearer, as well as its potential significance in the context of previous literature about circumglobal Rossby wave patterns (CRWPs). We also hope that the clarifications brought to several points of the manuscript made it even more understandable and interesting.

The most important modifications brought to the initial version are listed below:

- 1. A new Discussion section (Sec. 5) has been added to put the results of this study in the broader context of CRWP research, and to made explicit some limitations of the employed approach.
- 2. The Introduction section (Sec. 1) has been revised and extended: it features now an additional paragraph discussing the main theoretical and methodological issues related to the definition and detection of CRWPs.
- 3. Two extensive discussions about the interpolation procedure from wavenumber/frequency to wavenumber/phase speed periodograms have been added to the Supplement.
- 4. Fig.7 has been modified to display the median amplitude and phase of the MJO in the MJO phase diagram, merging the previous four plots into two (this comment did not result from reviewer's suggestions).

In the following, referee's comments are in black and our answers are below each of them, in **bold green**.

**Response to comment by Anonymous Referee #1**

**General Comments:**

This study explores the boreal winter midlatitude circulation features that emerge from the daily variability modes of the upper-tropospheric zonal wavenumber - phase speed spectrum. The authors focus on the origin and evolution of hemispheric-scale Rossby wave patterns that may characterize the first two modes and test hypotheses regarding the role of tropical convection, jet stream location, and low-level baroclinicity. The analyses are novel and valuable outcomes can be drawn from them. I have a few concerns regarding the basic rationale and scope of this study and some parts of the text require technical clarifications. I recommend a major revision of this manuscript and I hope that the specific remarks listed below will be helpful in this regard.

**Thanks for your comments, which have been very helpful to reflect about the significance of the results and the way they are presented, and for the time spent reading the manuscript. A detailed answer to each comment is given below.**

**Specific Comments:**

The main goal of this study is to investigate the origin and propagation of CRWPs. In some parts of the introduction and the conclusions, it appears that another objective is to showcase the potential of the two-dimensional spectral analysis on the uppertropospheric circulation. In the end, I am left wondering what is the exact scope of this study and what should be the take-home message for the readers. Perhaps, my confusion lies in the fact that the introduction (lines 14-67) touches on many themes and not all of them are directly relevant to the analyses. Could the authors clarify in the text the research questions adressed in this study and the respective conclusions?

Thanks for this comment, which allows us to reflect in a deeper way on how the employed methodology serves the scientific objectives of the study. Two aims are intertwined in the paper. The first aim of the study is to highlight how space/time spectral analysis can provide an independent assessment about the existence of CRWPs, without having to explicitly define such circulation patterns in advance and giving insights on their nature (l. 78-83). The success of this approach in identifying CRWPs leads to the second aim, that is to study some characteristics of their origin and propagation (l. 84-85). The introduction of the paper has been extended and made more precise by explicitly discussing open research questions in CRWP research (l. 54-74). Moreover, possible answers that results of this study can provide to these open questions are discussed in a new Discussion section (Sec. 5).

Following up on the previous comment, if the motivation of the study is to investigate and better understand CRWPs, then I find the employed methodology rather indirect. Instead of constructing daily n-Cp Fourier spectra of the midlatitudes, doing an EOF analysis on these spectra, and focusing on periods when the PC of the leading modes is large hoping that the composite patterns will exhibit long-lived and/or circumglobal wave patterns, why not directly diagnose CRWP events in the wind field by simply requiring that the RWP amplitude (section 2.2.2) is large enough in a large area for e.g. 5 consecutive days? The second option would result in composite maps that are easier to interpret and actually contain the wave patterns of interest. With the employed methodology, not all the observed CRWP events are necessarily included in the top PCs of EOF1 and EOF2, while these two leading modes may also contain days with no CRWPs, thus "contaminating" the lag-composite analyses with flow configurations beyond the authors' interest.

As correctly noticed, the adopted approach would not necessarily result in the identification of CRWPs. Instead, the principal component analysis is used to highlight the leading modes of variability of meridional wind amplitude in the wavenumber/phase speed spectral space, and it happens that the first two of these modes appear to project onto CRWPs. This result is not surprising: the EOF analysis provides the subsets of wavenumber/phase speed harmonics associated with the main modes of variability, and we also expect specific subsets of wavenumber/phase speed harmonics to be particularly enhanced during CRWPs (previous research, for instance, classified CRWPs with specific zonal wavenumbers). However, this implies that the discussed CRWPs are likely not the only ones, or that the same CRWP can emerge in different "flavors" in higher EOFs. These limitations and implicit assumptions are discussed in the new Discussion section (Sec.5).

The problem in using the simpler approach proposed by the reviewer is that the occurrence of large RWP amplitudes is not necessarily associated with CRWPs. Large RWP amplitudes could be associated with amplified Rossby waves occurring independently in the two storm tracks, while the basic idea of CRWP would involve a single wave pattern connecting different regions of the hemisphere. Also, we do not expect CRWPs to be necessarily associated with the most extreme RWP amplitudes over the midlatitudes: while there is of course some degree of meridional amplification of troughs and ridges, the really defining aspect of a CRWP is the (significant) zonal extent of its propagation, which is also harder to quantify.

On a more general note, the employed "indirect" approach has this big advantage: it allows to circumvent the challenge of objectively defining CRWPs before their identification. The only implicit assumption is that a CRWP would have a consistent spectral signature evaluated over several latitude circles, meaning that it would project on a defined range of wavenumbers and phase speeds as picked up by the spectral analysis. This allows to identify CRWPs without assuming a-priori how they look like, making the outcome less dependent on subjective choices. That's why we state in the paper that the employed approach lets CRWPs "emerge" from the spectral analysis (1.6). Previously employed CRWP diagnostics are affected by the significant difficulties in defining what Rossby waveguides are, difficulties also due to limits in the applicability of standard theoretical frameworks (e.g., the WKB approximation) for waves of significant meridional extent (as reviewed in White et al. 2021). The value of the approach adopted here is confirmed by the capability of retrieving large-scale circulation features that other studies discussed in association with the zonal propagation of Rossby waves. A relevant example, discussed in the abstract of the paper too (l. 9), becomes apparent in the enhanced meridional gradients of geopotential height evident during CRWP1 and CRWP2 events and that likely indicates an enhanced "waveguidability" (as in Manola et al. 2013, Wirth et al. 2020: see II.289-291). This series of advantages was not mentioned explicitly in the initial version of the paper, and is discussed comprehensively in the Introduction of the revised version (II. 54-74,75-82)

Given that the wavenumber-frequency spectra are based on the spectral coefficients (discrete Fourier Transform), writing down their equation would perhaps be more relevant than eq.1 (inverse discrete Fourier Transform). In addition, something seems to be off with the notation in eq.1. Since this is basically a 2D inverse DFT, I would expect the arguments of the spectral coefficient to be [n,j] rather than [n, $\omega_j$ ] (thus matching the two summation indices), the exponents to be like: exp[i2 $\pi(\lambda*n/N_L + t*j/N_T)$ ], and  $\omega_j$  to be equal to  $-2\pi j/N_T$ .

Thanks for this comment, as it gave us the opportunity to improve the clarity and the precision of the Methods section. In general, the choices behind the format and the style of the written formulas reflect the intention to highlight the planetary zonal wavenumber n and the frequency  $\omega_j$ , because we use these quantities to define the basis of wavenumber/frequency harmonics described in the methodology.

1) We decided not to change the formulation of the inverse Fourier transform to highlight that  $\hat{V}'$  are coefficient of the spectral decomposition of the input field V': however, additional informations about what  $\hat{V}'$  represent and how they can be used to obtain the spectral power density have been added in the text (ll.116, 119-121). 2) The longitude  $\lambda$  is actually expressed in radians, so the employed formulation with the planetary, a-dimensional wavenumber n should already be correct: this important information, previously missing, has been added to the revised version (l. 116) 3) Although the index j appears only as a subset  $\omega_j$ , we like the employed formulation because it emphasizes that what is in the exponent is actually a frequency, in units of cycles (6h)-1. 4) We would like to keep the minus sign in front of omega in the equation, because in this way the phase speed remains positive when waves are traveling eastward.

Is the methodology of constructing the wavenumber - phase speed spectra exactly the same as the one presented in Riboldi et al. (2019)? If so, this might be a useful remark for some readers. In any case, some things are not clearly described in the text. I am not sure what is the mathematical formulation behind the phrases: "smoothed 10 times in frequency using a three-point window" (line 103) and "The periodogram is interpolated along lines of constant phase speed" (line 104)? All the results are based on these steps, so it would greatly help the reader if they are described clearly with minimal jargon.

Thanks for this comment, the methodology to obtain the spectra involves a series of steps that are, indeed, not always simple to describe. We specified in the text that the procedure followed is exactly the same as in Riboldi et al. (2020), which was however based on meridional wind instead than on meridional wind anomalies. We changed the description of smoothing in frequency using simpler terms to "Each periodogram is smoothed 10 times in frequency with a 1-2-1 running average to further reduce noise" (II.123-124). Applications of this very simple filter are found, e.g., in Wheeler and Kiladis (1999) or Small et al. (2014). About the interpolation along lines of constant phase speed, the approach is the one documented by Randel and Held (1991) to transition from wavenumber/frequency spectra to wavenumber/phase speed spectra. A detailed explanation of the interpolation procedure and a schematic of the interpolation have been added to the Supplement of the paper (Text S1, Fig. S1). While working on this point, we noticed that we were occasionally referring to the wavenumber/phase speed periodograms as wavenumber/phase speed spectra. For further clarification, the symbol P has been added (1.121.125-126) to describe periodograms and distinguish them from the spectra (S).

Why do the days of top 15% PC1/PC2 (that is, up to 30% of all winter days) are included in the CRWP1/CRWP2 lists of events? Doesn't this result in too many events, given that the first 2 EOFs only explain 22.5% of the total variance? In this regard, it is perhaps no surprise that some days are found in both the CRWP1 and CRWP2 events (e.g. 16-24 January 1987, 25-26 December 2014) and that the typical duration of the events (as reported in the Supplement) is much larger that what the evolution in Fig.5 suggests. I presume that taking the top 5% or 10% PC days in each case would produce lists of less events but more indicative(?) of the respective EOF modes. Can the authors elaborate on their rationale behind the choice of this threshold and comment on the sensitivity of the results to its exact value?

The choice of thresholds always features a certain degree of subjectivity: in our case, we chose the upper and lower 15% and a duration of 5 days to have a sufficient balance between intensity of PC events and number of events. The results of the composite analysis, however, are not fundamentally different if the top 10% of PC values, or longer events, are chosen (cf. Fig. R1.1).